# ETA: Dual Evidence-Aware Uncertainty Learning for Open-Set Graph Domain Adaptation

## Abstract

Graph Neural Networks (GNNs) have shown great promise in node classification tasks, but their performance is often hindered by the scarcity of labeled nodes. Recently, graph domain adaptation has emerged as a promising solution to transfer knowledge from a labeled source graph to an unlabeled target graph. However, most existing methods typically rely on a closed-set assumption, which fails when unknown classes exist in the target domain. Toward this end, in this paper, we investigate the challenging open-set graph domain adaptation problem and propose a dual evidence-aware uncertainty learning framework ETA that simultaneously identifies unknown target nodes and enhances knowledge transfer under the evidential learning theory. Specifically, we adopt a dual-branch encoder to capture both implicit local structures and explicit global semantic consistency within the graph, and leverage evidential deep learning to integrate the evidence from both branches, where the resulting evidence is parameterized by a Dirichlet distribution to estimate class probabilities and enable uncertainty quantification. Based on the identified unknown target node, we further construct cross-domain neighborhoods and perform MixUp-based virtual sample generation in the latent space. Then, we introduce evidential adjacency-consistent uncertainty to evaluate uncertainty consistency across neighborhoods, which serves as auxiliary guidance for robust domain alignment. Extensive experiments on benchmark datasets demonstrate that ETA significantly outperforms state-of-the-art baselines in open-set graph domain adaptation tasks. Our code is available at anonymous.4open.science/r/ETA-FA1C/.

## 1 Introduction

Graph Neural Networks (GNNs) have emerged as a *de facto* paradigm for learning on graph-structured data, thanks to their powerful ability to capture both node-level features and relational dependencies through message passing mechanisms (Gilmer et al., 2017; Kipf & Welling, 2017), GNNs have achieved state-of-the-art results in numerous graph-based tasks (Wu et al., 2020b; Ju et al., 2024). In particular, node classification, which aims to predict the labels of nodes in a graph by jointly leveraging their features and topology information, has served as a fundamental task for a wide range of applications, including molecular property prediction (Guo et al., 2021; Zhuang et al., 2023), social behavior analysis (Liu et al., 2024; Wan et al., 2024), and cross-modal retrieval (Li et al., 2024). Nevertheless, the performance of GNNs largely hinges on the availability of abundant labeled data, which is often costly and time-consuming.

Graph transfer learning, which transfers knowledge from the well-annotated source graphs to an unlabeled target graph, has attracted increasing attention (Han et al., 2021; Zhu et al., 2021; 2024). Among various strategies in this paradigm, graph domain adaptation (GDA) has emerged as a key approach to mitigate distribution shifts between graphs, enabling the GNNs trained on the source domain graph to better adapt to the target domain graph (Qiao et al., 2023). Current efforts about GDA can be generally categorized into two main branches: Discrepancy-based methods aim to explicitly minimize the statistical divergence (i.e., Maximum Mean Discrepancy (MMD) (Shen et al., 2020) and graph subtree discrepancy (Wu et al., 2023)) between source and target domains for encouraging the alignment of their latent feature distributions. In contrast, adversarial-based methods use a domain discriminator to differentiate source and target graphs, generating indistinguishable node embeddings for domain alignment (Dai et al., 2022; Qiao et al., 2023).

Despite the effectiveness of these GDA methods, they typically focus on the closed-set assumption where all target domain nodes belong to one of the known classes from the source domain, which often does not hold true in a real-world scenario. In practice, target graphs may contain nodes from previously unseen classes, making it difficult for traditional GDA methods to generalize and leading to potential negative transfer (Wang et al., 2024; Yin et al., 2024). Towards this end, in this paper, we study the problem of *open-set graph domain adaptation*, which differs from closed-set adaptation by requiring the model to not only classify target nodes from known classes but also identify out-of-distribution (OOD) nodes belonging to unknown classes.

Actually, this problem is quite challenging due to the following twofold: (1) *Unknown Class Identification*. Traditional GDA approaches typically rely on pseudo-labeling to exploit unlabeled target data. However, under open-set scenarios, these methods often fail to distinguish unknown target nodes from known ones, resulting in incorrect label assignments that propagate errors and ultimately hinder effective knowledge transfer. (2) *Domain Alignment under Inadequate Supervision* The presence of unknown classes exacerbates the label scarcity issue in the target domain, making it challenging to establish reliable alignment between source and target domains. Existing GDA methods typically aim to learn domain-invariant representations through global feature alignment, yet these approaches often overlook the semantic discrepancies introduced by unknown classes. Besides, prior work (Wang et al., 2024) predominantly relies on predictive entropy for identifying unknown classes, which may vary substantially across different open-set GDA scenarios, thereby leading to unstable performance.

In this paper, we propose ETA illustrated as Figure 1, a novel Dual **E**vidence-Aware Uncertainty Learning framework for Open-Se**T** graph domain **A**daption (ETA), which identifies the unknown nodes and facilitates knowledge transfer from labeled source graph to the unlabeled target graph under the evidential learning theory (Sensoy et al., 2018). Specifically, our ETA incorporates a dual-branch architecture composed of an edge-oriented encoder and a path-oriented encoder to fully leverage the complementary information encoded in node attributes and graph topology. The edge-oriented encoder implicitly captures local topological semantics through message passing over immediate node neighborhoods, while the path-oriented encoder explicitly aggregates information across diverse relational paths. Then, to reliably identify unknown classes, we introduce an evidence-aware classification module to facilitate uncertainty quantification, enabling stable identification of unknown target nodes based on the principles of evidential learning. Furthermore, to mitigate domain shift under the open-set scenario, we construct cross-domain neighborhoods by retrieving the $k$-nearest neighbors across domains. For each node, we perform latent space MixUp with its cross-domain neighbors to generate informative virtual samples. To quantify the alignment reliability, we introduce evidential adjacency-consistent uncertainty estimation, which assesses the consistency of uncertainty across different adjacency sets. Based on this, we provide auxiliary supervision and promote more robust cross-domain alignment. Extensive experiments are conducted on several benchmark datasets to evaluate the performance of our proposed ETA, and the results highlight the superiority of the framework for open-set graph domain adaptation.

## 2 METHODOLOGY

### 2.1 PROBLEM DEFINITION

**Source Domain Graph.** Let the source domain graph be denoted as $\mathcal{G}^s = \{\mathcal{V}^s, \mathcal{E}^s, \boldsymbol{X}^s, \boldsymbol{Y}^s\}$, where $\mathcal{V}^s$ and $\mathcal{E}^s$ represents the node and edge set respectively. Each node $v \in \mathcal{V}^s$ is associated with a $d$-dimensional attribute vector, and the collective node features are represented by the matrix $\boldsymbol{X}^s \in \mathbb{R}^{|\mathcal{V}^s| \times d}$. The structure of the graph can be characterized by the adjacency matrix $\boldsymbol{A}^s \in \{0, 1\}^{|\mathcal{V}^s| \times |\mathcal{V}^s|}$, where $\boldsymbol{A}^s_{ij} = 1$ if there is an edge $(v_i, v_j) \in \mathcal{V}_s$ between node $v_i$ and $v_j$, otherwise $\boldsymbol{A}^s_{ij} = 0$. The corresponding degree matrix $\boldsymbol{D} \in \mathbb{R}^{|\mathcal{V}^s| \times |\mathcal{V}^s|}$ is diagonal, with each entry $\boldsymbol{D}_{ii} = \sum_{j=1}^{|\mathcal{V}^s|} \boldsymbol{A}^s_{ij}$ indicating the degree of node $v_i$. We denote the label matrix as $\boldsymbol{Y}^s \in \mathbb{R}^{|\mathcal{V}^s| \times |\mathcal{C}_s|}$, where each node label $\boldsymbol{y}_v$ corresponds to one of the $|\mathcal{C}_s|$ classes in the source domain label space $\mathcal{C}_s$.

**Target Domain Graph.** Similarly, the target domain graph is denoted as $\mathcal{G}^t = \{\mathcal{V}^t, \mathcal{E}^t, \boldsymbol{X}^t\}$ with completely unlabeled node set $\mathcal{V}^t$ and edge set $\mathcal{E}^t$. The node feature and adjacency matrix are denoted as $\boldsymbol{X}^t \in \mathbb{R}^{|\mathcal{V}^t| \times d}$ and $\boldsymbol{A}^t \in \mathbb{R}^{|\mathcal{V}^t| \times |\mathcal{V}^t|}$. Let $\mathcal{C}_t$ represent the label space of the target domain graph. To facilitate alignment, we construct a unified feature space across the source and target domains.

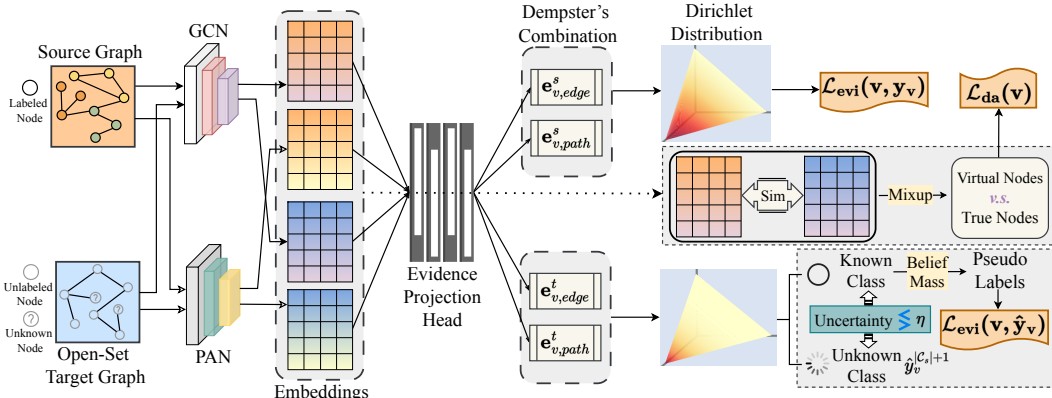

Figure 1: Illustration of the proposed framework ETA, it consists of three components as follows. (1) *Dual-Branch Encoder*, which jointly captures local structural and global semantic information. (2) *Evidence-Aware Classifier*, which employs a Dirichlet-based evidential framework to quantify uncertainty and identify unknown nodes. (3) *Evidential Adjacency-MixUp*, which constructs cross-domain neighborhoods and performs consistency-aware MixUp in latent space for robust alignment.

**Open-Set Graph Domain Adaptation.** We consider the open-set graph domain adaptation (OSGDA) problem, which involves a labeled source graph $\mathcal{G}^s$ and an unlabeled target graph $\mathcal{G}^t$. Unlike conventional GDA settings that assume identical label spaces across domains, the open-set scenario allows for the presence of unknown classes exclusive to the target domain, such that $\mathcal{C}_s \subset \mathcal{C}_t$. We denote the shared class set as $\mathcal{C}_s = \mathcal{C}_s \cap \mathcal{C}_t$ and the unknown class set as $\bar{\mathcal{C}}_t = \mathcal{C}_t \setminus \mathcal{C}_s$. The objective is to transfer knowledge from the source domain to correctly classify target nodes into $|\mathcal{C}_s| + 1$ classes with nodes from $\bar{\mathcal{C}}_t$ as an additional unknown class. The predictive model is formalized as $f = h(g(\boldsymbol{x}_v))$, where $g(\cdot)$ extracts features and $h(\cdot)$ performs classification.

## 2.2 DUAL-BRANCH ENCODER FOR CONSISTENCY DELVING

To comprehensively capture both node-centric local and high-order global consistency relationships within the graph, we construct a dual-branch encoder from two complementary perspectives, consisting of an implicit edge-oriented branch and an explicit global path-oriented branch, thus enrich the semantic aggregation from multiple perspectives.

**Edge-Oriented Graph Encoder.** Given the adjacency matrix $\boldsymbol{A}^*$ and node feature matrix $\boldsymbol{X}^*$ ($* \in \{s, t\}$) of the source and target domain graphs, the edge-oriented branch is designed to capture local structural patterns by aggregating information from immediate neighbors. Specifically, we utilize a message-passing scheme to encode local consistency knowledge (i.e., neighboring nodes are more likely to share the same label) in an implicit manner. The update rule can be:

$$\boldsymbol{Z}_{edge}^{*,(l)} = \sigma(\tilde{\boldsymbol{D}}^{*^{-1/2}} \tilde{\boldsymbol{A}}^* \tilde{\boldsymbol{D}}^{*^{-1/2}} \boldsymbol{Z}_{edge}^{*,(l-1)} \boldsymbol{W}_{edge}^{(l)}), \tag{1}$$

where $\boldsymbol{Z}_{edge}^{*,(l)}$ and $\boldsymbol{W}_{edge}^{(l)}$ denote the node embeddings and filter weight at $l$-th layer. $\tilde{\boldsymbol{A}} = \boldsymbol{A} + \boldsymbol{I}$ and $\tilde{\boldsymbol{D}}$ is the corresponding degree matrix. By stacking $L$ layers, the encoder gradually enlarges each node's receptive field, enabling the model to capture local topological dependencies $\boldsymbol{Z}_{edge}^{*,(L)}$.

**Path-Oriented Graph Encoder.** While the edge-oriented branch effectively encodes local consistency knowledge, it remains limited in modeling high-order structural dependencies inherent in graph data (Ma et al., 2020; Wang et al., 2025). To address this, we incorporate a path-oriented encoder that explicitly captures global semantics through multi-hop paths. Instead of relying solely on immediate neighbors, this encoder aggregates information along multiple paths between node pairs thus emphasizing long-range dependencies, The global consistency knowledge can be formally updated as:

$$\boldsymbol{Z}_{path}^{*,(l)} = \sigma\Big(\boldsymbol{M}^{*^{-1/2}} \sum_{p=0}^{P} e^{-\frac{E_p}{\tau}} \boldsymbol{A}^{*p} \boldsymbol{M}^{*^{-1/2}} \boldsymbol{Z}_{path}^{*,(l-1)} \boldsymbol{W}_{path}^{(l)}\Big), \tag{2}$$

where $\boldsymbol{M}^*$ denotes the normalization matrix and $\boldsymbol{A}^{*p}$ denotes the $p$-th power of adjacency matrix, capturing $p$-hop connectivity among nodes. By aggregating over multi-hop paths with learnable path weight $e^{-\frac{E_p}{\tau}}$, the encoder explicitly integrates high-order topological dependencies based on their structural importance. Here, $E_p$ denotes the energy assigned to the $p$-length path, $\tau$ is a temperature parameter, and similarly, $\boldsymbol{W}_{path}^{(l)}$ is the weight matrix at the $l$-th layer. We can also stack $L$ path convolutional layers and the global topological dependencies can be $\boldsymbol{Z}_{path}^{*,(L)}$.

## 2.3 EVIDENCE-AWARE CLASSIFIER WITH UNCERTAINTY QUANTITATION

Given the encoded consistency knowledge from two branches, we employ evidential deep learning (Sensoy et al., 2018) (EDL) to quantify classification uncertainty by modeling both the likelihood of each class and the overall uncertainty in the prediction to identify unknown nodes in target graph.

**Evidence-Aware Uncertainty Prediction.** The Dempster–Shafer Theory of Evidence (DST) extends the traditional Bayesian approach by incorporating subjective probabilities (Dempster, 1968), which assigns belief masses to the set of all possible states. Subjective Logic (SL) (Jsang, 2018) formalizes DST using a Dirichlet distribution (defined in Appendix A.1). In multi-class classification, these belief masses can be distributed across both known and potentially unknown classes. By assigning a portion of the belief mass to the entire frame, the model expresses uncertainty by indicating that the true class is unknown (Sensoy et al., 2018). Specifically, for each node $v$ in the graph, we consider a frame of $|\mathcal{C}_t|$ mutually exclusive singletons (class labels), where the model assigns a belief mass $b_v^k$ to each class $k \in \mathcal{C}_s$, along with an overall uncertainty mass $u_v \in \bar{\mathcal{C}}_t$ reflecting insufficient evidence for known classes. Accordingly, these masses satisfy the constraint:

$$u_v + \sum\nolimits_{k=1}^{|\mathcal{C}_s|} b_v^k = 1. \tag{3}$$

Let $\boldsymbol{e}_v = [e_v^1, \ldots, e_v^K]$ be the *evidence* vectors, with each element $e_v^k \geq 0$ corresponds to the evidence of $k$-th class. The parameters of the Dirichlet distribution $\boldsymbol{\alpha}_v = [\alpha_v^1, \ldots, \alpha_v^K]$ can be induced from $\boldsymbol{e}_v$, i.e., $\boldsymbol{\alpha}_v = \boldsymbol{e}_v + 1$. Then, the belief and the uncertainty mass can be calculated as:

$$b_v^k = \frac{e_v^k}{S_v} = \frac{\alpha_v^k - 1}{S_v}, u_v = \frac{K}{S_v}, \tag{4}$$

where $S_v = \sum_{k=1}^{K}(e_v^k + 1) = \sum_{k=1}^{K} \alpha_v^k$ denotes the Dirichlet strength, reflecting the total amount of evidence. Intuitively, a higher $e_v^k$ leads to a greater belief mass $b_v^k$, indicating stronger support for class $k$, while a smaller total evidence results in higher uncertainty $u_v$, representing limited confidence in all classes. Therefore, we leverage the extracted consistency knowledge of each node to construct corresponding multinomial opinions. Taking the local consistency knowledge as an example, the evidence vector is computed as $\boldsymbol{e}_{v,edge}^* = h(\boldsymbol{Z}_{v,path}^{*,(L)})$, where $h(\cdot)$ is a fully connected evidence projection head followed by a non-negative activation. Accordingly, the parameters of the Dirichlet distribution can be expressed as $\boldsymbol{\alpha}_{v,edge}^* = \boldsymbol{e}_{v,edge}^* + 1$.

**Dempster's Rule of Combination.** Considering that the presence of unknown classes in OSGDA introduces noise, the knowledge obtained from the two complementary perspectives may exhibit certain conflicts. To address this, we adopt the principled framework provided by the Dempster-Shafer theory to fuse the evidence from both branches, enabling the model to derive a more comprehensive and reliable representation of the underlying knowledge. The core principle of the rule is to retain only the parts where both branches provide consistent support and treat the inconsistent portions as conflict mass, which is subsequently normalized. And the detailed definition is in Appendix A.2

Based on the Dempster's Rule of combination, the joint evidence for the node $v$ and corresponding parameters of the Dirichlet distribution from two complementary perspectives can be induced as:

$$S_v = \frac{K}{u_v}, e_v^k = b_v^k \times S_v, \text{and } \alpha_v^k = e_v^k + 1. \tag{5}$$

The predictive opinion probabilities $\boldsymbol{p}_v = \{p_v^1, \ldots, p_v^K\}$ are obtained by the mean of the Dirichlet distribution, namely probability of the $k$-th singleton can be $p_v^k = \frac{\alpha_v^k}{S_v}$. Note that we generate the uncertainty mass for the nodes in target graph and quantify the node with an unknown class label as:

$$\hat{y}_v^{|\mathcal{C}_s|+1} = \begin{cases} 1, & \text{if } u_v > \eta, \\ 0, & \text{otherwise} \end{cases}, \tag{6}$$

where $\eta$ is the threshold. To investigate whether the value of $u_v$ can effectively distinguish OOD samples from known classes in OSGDA tasks, we conduct a theoretical analysis of the properties of $u_v$ in EDL and present the following theorem.

**Theorem 1.** *In the graph node-level classification task, let $g : \boldsymbol{X} \to \boldsymbol{Z} \subset \mathbb{R}^d$ be the GNN feature extractor, and let $h : \boldsymbol{Z} \to \mathbb{R}_{\geq 0}^K$ be the evidence projection head. According to EDL, we have following evidence input of node $v$ and corresponding parameters of Dirichlet distribution:*

$$e_v = h(g(X_v)) \in \mathbb{R}_{\geq 0}^K, \alpha_v = e_v + \mathbf{1}, S_v = \sum_{k=1}^K \alpha_v^k = \sum_{k=1}^K e_v^k + K, u_v = \frac{K}{S_v}. \quad (7)$$

*Then, for any sample $v$ satisfying $\min_{c \in \mathcal{C}_s} \|Z_v - m_c\| \geq d$, we have*

$$S_v \leq G(d), u_v = \frac{K}{S_v} \geq \frac{K}{G(d)}, \quad (8)$$

*where $G(d) := K + \max_{c \in \mathcal{C}_s} \sum_{k=1}^K \varphi_{c,k}(d), d \geq 0, \varphi_{c,k}(\cdot)$ is a non-negative, non-increasing function, and $m_c$ is a prototype. Since $G(d)$ is nonincreasing in $d$, the lower bound $\frac{K}{G(d)}$ is nondecreasing in $d$. Thus, the farther a sample is from all known prototypes, the larger its uncertainty is guaranteed to be.*

The proof is in Appendix B.1. Theorem 1 guarantees that the $u_v$ is highly related to the distance in the embedding space, thus for the OOD samples which are far way from the known prototypes in the embedding space, we can set a threshold to distinguish them from the known classes, which is reasonable in the OSGDA tasks. For other nodes with known class labels, we generate the pseudo label $\hat{\boldsymbol{y}}_v$ by assigning 1 to the class with the highest belief mass $\hat{y}_v^k = \arg\max_k p_v^k$.

**Learning to Form Opinions.** We design and train neural networks to express their predictive opinions as Dirichlet distributions. For each node $v$, the network estimates the non-negative evidence vector $\boldsymbol{e}_v$, based on which the Dirichlet parameters are computed as $\boldsymbol{\alpha}_v = \boldsymbol{e}_v + 1$. We treat the Dirichlet distribution $D(\boldsymbol{p}_v|\boldsymbol{\alpha}_v)$ as a prior on the multinomial likelihood $\text{Mult}(\boldsymbol{y}_v|\boldsymbol{p}_v)$, and the loss function can be formulated as:

$$\mathcal{L}_{ace}(v, \boldsymbol{y}_v) = \int \|y_v - p_v\|_2^2 \frac{1}{B(\boldsymbol{\alpha}_v)} \prod_{k=1}^K (p_v^k)^{\alpha_v^k - 1} d\boldsymbol{p}_v = \sum_{k=1}^K \left((y_v^k)^2 - 2y_v^k \mathbb{E}[p_v^k] + \mathbb{E}[(p_v^k)^2]\right). \quad (9)$$

According to EDL, the above loss function can be rewritten in a more interpretable form as:

$$\mathcal{L}_{ace}(v, \boldsymbol{y}_v) = \sum_{k=1}^K \underbrace{(y_v^k - \frac{\alpha_v^k}{S_v})^2}_{(\mathcal{L}_v^k)^{err}} + \underbrace{\frac{\alpha_v(S_v - \alpha_v)}{S_v^2(S_v + 1)}}_{(\mathcal{L}_v^k)^{var}}. \quad (10)$$

And we have following three propositions that present the properties of the loss function above:

**Proposition 1.** *For any $\alpha_v^k \geq 1$, the inequality $(\mathcal{L}_v^k)^{var} < (\mathcal{L}_v^k)^{err}$ satisfied.*

**Proposition 2.** *For a given sample $v$ with the correct label $k$, $L_v^{err}$ decreaces when new evidence is added to $\alpha_v^k$ and increases when evidence is removed from $\alpha_v^k$.*

**Proposition 3.** *For a given sample $v$ with the correct class label $j$, $L_v^{err}$ decreases when some evidence is removed from the biggest Dirichlet parameter $\alpha_v^l$ such that $l \neq j$.*

The proofs of the propositions are in Appendix B.2. The propositions gaurantee the above loss function can effectively generate more evidence for the known samples while reverting to high uncertainty when encountering incomprehensible samples (e.g., OOD samples), thereby achieving strong alignment with the OSGDA tasks. Besides, we further encourage the model to generate less evidence for incorrect classes by introducing the following KL divergence term:

$$\mathcal{L}_{kl}(v, \boldsymbol{y}_v) = KL[D(\boldsymbol{p}_v|\tilde{\boldsymbol{\alpha}}_v)\|D(\boldsymbol{p}_v|\mathbf{1}]$$

$$= \log\left(\frac{\Gamma(\sum_{k=1}^K \tilde{\alpha}_v^k)}{\Gamma(K)\prod_{k=1}^K \Gamma(\tilde{\alpha}_v^k)}\right) + \sum_{k=1}^K (\tilde{\alpha}_v^k - 1)\left[\psi(\tilde{\alpha}_v^k) - \psi\left(\sum_{k'=1}^K \tilde{\alpha}_v^{k'}\right)\right], \quad (11)$$

where $\psi(\cdot)$ denotes the digamma function, and $\tilde{\boldsymbol{\alpha}}_v = \boldsymbol{y}_v + (1 - \boldsymbol{y}_v) \odot \boldsymbol{\alpha}_v$ is a modified Dirichlet parameter that preserves the evidence for the ground-truth class, ensuring them not mistakenly shrink to 0. The evidence-aware loss can be:

$$\mathcal{L}_{evi}(v, \boldsymbol{y}_v) = \mathcal{L}_{ace}(v, \boldsymbol{y}_v) + \lambda \mathcal{L}_{kl}(v, \boldsymbol{y}_v), \tag{12}$$

where $\lambda$ is the balance factor to adjust the impact of the regularization trem.

## 2.4 EVIDENTIAL ADJACENCY-MIXUP FOR DOMAIN ALIGNMENT

Despite the generation of pseudo labels, the challenge of severe domain shift remains, which may lead to unreliable supervision signals. To mitigate this, we introduce an evidential adjacency-MixUp to provide auxiliary supervision for domain alignment.

**Evidential Domain Adjacency-MixUp.** We identify a cross-domain neighborhood to facilitate knowledge transfer and promote semantic consistency across domains. In detail, we take the edge-oriented branch as an example, and retrieve $k$ mutual nearest cross-domain neighbors for node $v \in \mathcal{V} = \mathcal{V}_s \cup \mathcal{V}_t \backslash \{v' | \hat{y}_{v'}^{|\mathcal{C}_s|+1} = 1\}$. Then, for each node sample, we take a combination of all neighbors as the virtual MixUp informative virtual samples $v'$, defined as:

$$\boldsymbol{Z}_{v',edge}^{(L)} = \sum\nolimits_{u \in \mathcal{T}(v)} \lambda_v^u \boldsymbol{Z}_{u,edge}^{*,(L)}, \tag{13}$$

where $\mathcal{T}(v)$ is the cross-domain neighbors. $\lambda_v^u = s(\boldsymbol{Z}_{v,edge}^{*',(L)}, \boldsymbol{Z}_{u,edge}^{*,(L)}) / \sum_{u'} s(\boldsymbol{Z}_{v,edge}^{*',(L)}, \boldsymbol{Z}_{u',edge}^{*,(L)})$ is the MixUp weight with $* = t, *' = s$ for $v \in \mathcal{V}_s$, otherwise $* = s, *' = t$; $s(\cdot)$ denotes cosine similarity.

**Adjacency-Consistent Uncertainty for Domain Alignment.** To select samples with minimal noise and ensure that the transferred knowledge occurs with high confidence between instances of the same class across domains, we propose a strategy that integrates both individual and interaction-based evidence characteristics to quantify the adjacency-consistent uncertainty. Specifically, we define two components: (1) an individual term *Ind*, which captures evidential characteristics of a virtual node $v'$ through the maximum Dirichlet parameter $\max_k(\boldsymbol{\alpha}_{v'})$ and the residual $S_{v'} - \max_k(\boldsymbol{\alpha}_{v'})$, and (2) an interaction term *Int*, which models the discrepancy between a node and its neighbors. Then, we maximize the evidential consistency between node $v$ and its virtual sample $v'$, which can be defined as:

$$\mathcal{L}_{da}(v) = Ind \cdot Int, \quad \text{where } Ind = \log\left(\frac{S_{v'} - \max_k(\boldsymbol{\alpha}_{v'})}{\max_k(\boldsymbol{\alpha}_{v'})}\right), Int = \left\|\frac{\boldsymbol{\alpha}_v}{S_v} - \frac{\boldsymbol{\alpha}_{v'}}{S_{v'}}\right\|_1. \tag{14}$$

## 2.5 FRAMEWORK SUMMARIZATION

The final objective consists of the evidence-aware and auxiliary consistency loss on the source domain graph and the target domain graph with the known pseudo label, summarized as:

$$\mathcal{L} = \sum\nolimits_{v \in \mathcal{V}_s} \mathcal{L}_{evi}(v, \boldsymbol{y}_v) + \sum\nolimits_{v \in \mathcal{V} \backslash \mathcal{V}_s} \mathcal{L}_{evi}(v, \hat{\boldsymbol{y}}_v) + \sum\nolimits_{v \in \mathcal{V}} \mathcal{L}_{da}(v) \tag{15}$$

**Time Complexity.** Assume the number of nodes and edges in the source and target domains are $\mathcal{V}^s, \mathcal{V}^t, \mathcal{E}^s, \mathcal{E}^t$, respectively. We adopt GCN and PAN as the backbones of the two branches. The time complexity for the embedding stage is $\mathcal{O}(\mathcal{V}^s + \mathcal{V}^t + \mathcal{E}^s + \mathcal{E}^t)$. The evidence fusion process has a time complexity of $\mathcal{O}(\mathcal{V}^s + \mathcal{V}^t)$, and computing the evidence loss for both source and target domains also requires $\mathcal{O}(\mathcal{V}^s + \mathcal{V}^t)$. The time complexity of domain alignment is $\mathcal{O}(\mathcal{V}^s \cdot \mathcal{V}^t)$. In practice, to reduce computational cost, we select anchor nodes from the source domain for cross-domain alignment. Let the number of anchor nodes be $c$; then, this part has a time complexity of $\mathcal{O}(c \cdot \mathcal{V}^t)$. Thus, the overall time complexity is: $\mathcal{O}(\mathcal{E}^s + \mathcal{E}^t + \mathcal{V}^s + \mathcal{V}^t + c \cdot \mathcal{V}^t)$.

# 3 EXPERIMENT

## 3.1 EXPERIMENT SETTINGS

**Datasets.** Our experiments involve three widely used citation network datasets from (Tang et al., 2008): ACMv9 (A), Citationv1 (C), and DBLPv7 (D), and we follow the data preprocessing procedures proposed by (Qiao et al., 2023). For the OSGDA task using these three citation networks, we

Table 1: Performance of various methods across six open-set domain adaptation tasks (ACC (%) and HS (%) ). The best performance is marked in **bold**, and the second-best is underlined.

| Methods | A⇒D | | D⇒A | | A⇒C | | C⇒A | | C⇒D | | D⇒C | | Average | |
|---|---|---|---|---|---|---|---|---|---|---|---|---|---|---|
| | ACC | HS | ACC | HS | ACC | HS | ACC | HS | ACC | HS | ACC | HS | ACC | HS |
| GCN | 45.10 | 41.80 | 38.95 | 39.52 | 46.36 | 43.91 | 44.14 | 43.66 | 48.45 | 44.61 | 42.26 | 41.25 | 44.21 | 42.46 |
| GraphSAGE | 48.26 | 46.22 | 43.14 | 42.84 | 50.60 | 49.04 | 48.13 | 46.36 | 51.72 | 48.66 | 47.20 | 46.70 | 48.17 | 46.64 |
| DANN | 33.30 | 28.07 | 34.58 | 36.53 | 39.64 | 41.22 | 34.47 | 34.42 | 36.92 | 41.88 | 35.20 | 35.46 | 35.68 | 34.16 |
| CDAN | 31.13 | 21.65 | 29.03 | 27.76 | 30.99 | 26.00 | 31.72 | 30.91 | 35.69 | 30.47 | 28.62 | 21.82 | 31.20 | 26.44 |
| OSBP | 28.56 | 11.27 | 26.20 | 12.91 | 29.32 | 11.15 | 27.80 | 7.34 | 33.81 | 18.89 | 28.63 | 14.16 | 29.05 | 12.62 |
| DANCE | 60.54 | 25.99 | 53.27 | 39.53 | 63.23 | 39.15 | 60.44 | 35.88 | 64.29 | 28.98 | 57.62 | 39.50 | 59.90 | 34.84 |
| UDAGCN | 36.20 | 26.59 | 31.90 | 12.31 | 37.44 | 32.01 | 35.64 | 22.76 | 41.88 | 36.48 | 35.50 | 25.09 | 36.43 | 25.87 |
| ASN | 56.40 | 37.55 | 47.49 | 43.93 | 59.88 | 49.82 | 57.51 | 47.87 | 56.65 | 45.62 | 56.97 | 46.19 | 55.81 | 45.16 |
| SDA | 61.60 | 49.22 | 51.36 | 50.86 | **64.47** | 55.27 | 61.67 | 55.89 | **67.51** | 55.35 | 57.74 | 54.72 | 60.73 | 53.55 |
| UAGA | 58.44 | 56.28 | 52.97 | 47.40 | 64.13 | 53.66 | 55.17 | 54.17 | 62.18 | 61.17 | **62.58** | **58.00** | 59.25 | 55.11 |
| ETA | **61.87** | **60.96** | **54.21** | **53.76** | 62.76 | **62.57** | **61.82** | **58.17** | 64.78 | **63.50** | 59.96 | 55.90 | **60.90** | **59.15** |

sequentially select one as the source domain and the other two as target domains. In each setting, two classes are removed from the source domain as unknown classes, while the remaining three classes are treated as known classes for the experiments. More details are provided in Appendix F.

**Baselines and evaluation criteria.** We select several classical GNN models as well as state-of-the-art methods in open-set and domain adaptation tasks as baselines for comparison: 1) GCN (Kipf & Welling, 2017) and GraphSAGE (Hamilton et al., 2017); 2) DANN (Ganin et al., 2016) and CDAN (Long et al., 2018); 3) OSBP (Saito et al., 2018) and DANCE (Saito et al., 2020); 4) UDAGCN (Wu et al., 2020a) and ASN (Zhang et al., 2021); 5) SDA (Wang et al., 2024) and UAGA (Shen et al., 2025). More details about the datasets are provided in Appendix G. To evaluate the performance of different methods on the OSGDA task, we adopt four metrics as evaluation criteria: average class accuracy on known classes ($ACC_k$), accuracy on unknown classes ($ACC_u$), average per-class accuracy over the entire domain (ACC), and the H-score (HS) (Fu et al., 2020).

**Implementation details.** We adopt a 2-layer GCN (Kipf & Welling, 2017) and PAN (Ma et al., 2020) as the backbones for our two branches, with the feature embedding dimension of both networks set to 512. The hyperparameters are configured as: $\eta$=0.65, $\lambda$=0.5, $k$=3, learning rate=0.005, and weight decay=0.001. Across the six domain adaptation tasks constructed from the three datasets, we randomly select two out of the five labels as unknown classes for each task and train the model for 200 epochs. Each task is repeated 10 times, and we record average performance as final results.

## 3.2 PERFORMANCE AND DISCUSSION

To verify the superiority of our method, Table 1 record the ACC and HS of our ETA and competitive baselines. Overall, our ETA consistently achieves the best performance, followed by SDA and UAGA. Classical GNNs perform moderately, while OSBP, CDAN, and DANN perform worst, indicating that closed-set and open-set graph domain adaptation methods perform better than unsupervised domain adaptation approaches in visions. We speculate that in open-set scenarios, unknown nodes would cause disruption to known

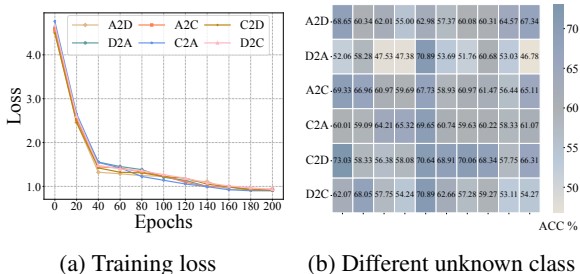

(a) Training loss   (b) Different unknown class

Figure 2: Convergence analysis and impact of different unknown class under different setups.

node classification, making cross-domain transfer even more difficult. SDA amd UAGA performs almost better than other baselines, showing their effectiveness for OSGDA tasks. However, while SDA and UAGA achieves competitive ACC, their HS is relatively low, indicating imbalance between known and unknown classes. In contrast, our ETA attains higher ACC in nearly all OSGDA tasks and significantly outperforms SDA amd UAGA in HS (average improvement of 4.04%), showing our model could learn more balanced semantic features, thus demonstrating our approach's superiority. Figure 2a shows the loss curve of our ETA during training, illustrating rapid convergence. Figure 2b shows the impact on performance when different pairs of labeled classes are removed as unknown

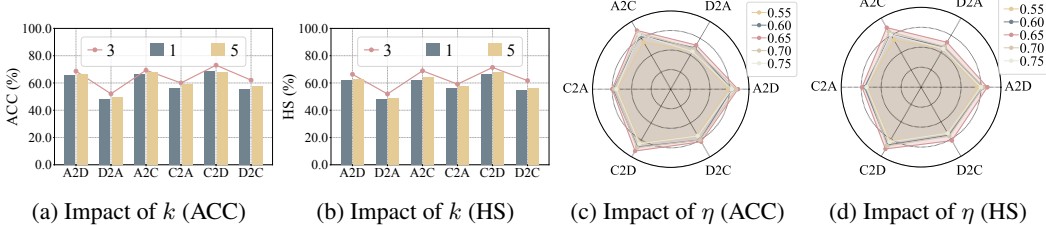

(a) Impact of $k$ (ACC)  (b) Impact of $k$ (HS)  (c) Impact of $\eta$ (ACC)  (d) Impact of $\eta$ (HS)

Figure 4: Sensitivity analysis of two key hyperparameters.

classes (e.g., every column indicates a different exoeriment with different unknown classes), where ETA consistently achieves stable, balanced performance across different experiments, demonstrating strong robustness of our ETA. The detailed results and analysis are supplied in Appendix C.

### 3.3 ABLATION STUDY AND SENSITIVITY ANALYSIS

**1) Ablation Study.** We conduct ablation studies to investigate the effectiveness of each component in our method, with partial results shown in Figure 3. It can be observed that removing any component leads to a certain degree of performance degradation. In particular, the most significant drops occur when removing target domain evidence loss (w/o $\mathcal{L}_{evi}$) and domain align-

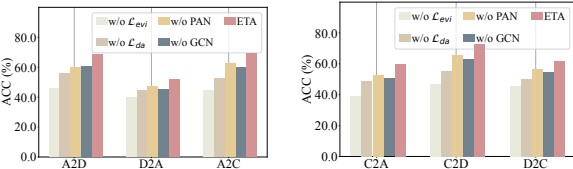

Figure 3: Ablation analysis (ACC).

ment (w/o $\mathcal{L}_{da}$), with the removal of evidence loss causing a more pronounced decline. This highlights the importance of leveraging evidence to distinguish unknown-class nodes in order to prevent interference with the training of known classes, as well as the necessity of selecting high-confidence consistent neighbors for effective domain adaptation. In addition, replacing the dual-branch structure with a single-branch one (w/o PAN and w/o GCN) also results in performance degradation, indicating that the fused evidence from multiple branches is more reliable than that from a single branch, which enhances the model's ability to accurately characterize node classification, and plays a more trustworthy role in distinguishing known and unknown nodes in the OSGDA tasks.

**2) Sensitivity Analysis.** To investigate the impact of different values of hyperparameters of our ETA, we conduct a series of hyperparameter analysis as follows.

**Effect of $k$.** We study the range of $k \in \{1, 3, 5\}$ which controls the number of neighbors involved in the domain alignment, and the results are presented in Figures 4a and 4b. As can be observed, the performance first improves and then declines as $k$ increases. When $k$ is too small, few neighbor samples are incorporated, resulting in insufficient cross-domain alignment due to limited sample diversity. Conversely, when $k$ is too large, the probability of including inconsistent neighbors rises, thereby introducing noise that disrupts the adaptation process and reduces performance. To strike a balance between sufficient information and minimal noise, we finally set $k = 3$ for our approach.

**Effect of $\eta$.** Here we study $\eta$ range from $\{0.55, 0.60, 0.65, 0.70, 0.75\}$, which distinguishes unknown-class nodes based on node uncertainty, and the results are presented in Figures 4c and 4d. It shows that as $\eta$ increases, performance first rises and then declines. When $\eta$ is too small, many known-class nodes are misclassified as unknown, weakening the model's ability to identify unknown nodes and reducing performance. Conversely, a large $\eta$ causes too many unknown nodes to be treated as known, introducing noise into training and degrading performance. To achieve a balanced threshold, we set $\eta = 0.65$ to maintain a clear decision boundary between known and unknown classes. Besides, we adopt a dynamically evolving thresholding strategy for our method: the threshold is first initialized with a predefined value, and then gradually increased as training progresses. This dynamic scheme enables the model to incorporate a relatively large set of unknown class samples during the early phase of training, while focusing on fewer but more reliable unknown samples in later stages. Such progression helps our method better adapt to a wide range of OSGDA scenarios.

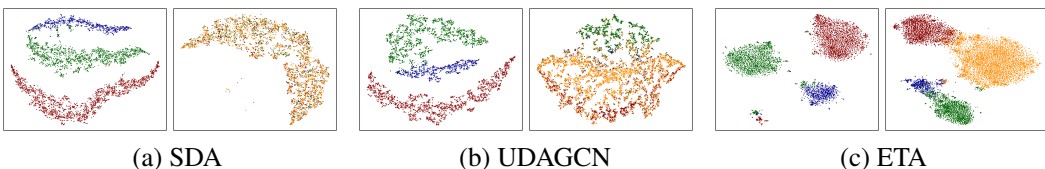

(a) SDA          (b) UDAGCN          (c) ETA

Figure 5: T-SNE visualization of source and target domain node embedding. Each pair, left: ACMv9 (source domain) and right: Citationv1 (target domain), corresponds to one method.

### 3.4 VISUALIZATION ANALYSIS

To further demonstrate the effectiveness of our ETA in domain adaptation, we perform t-SNE visualization on the node representations learned by our ETA and two competitive baselines. The results are shown in Figure 5. It is evident that in the source domain, all three methods produce clear separations among different classes, indicating an effective representation learning. However, in the target domain, the node representations learned by the two baselines lack clear class boundaries, suggesting that the presence of unknown classes in the open-set setting disrupt their ability to learn effective representations. In contrast, our ETA yields well-separated clusters even in the target domain, which clearly demonstrates its ability to capture domain-invariant and discriminative node representations, thereby enabling more effective cross-domain knowledge transfer.

### 3.5 EFFECTIVE DIFFERENTIATION BETWEEN KNOWN AND UNKNOWN CLASSES

To further highlight the advantages of our method over existing approaches on the OSGDA task, we record the accuracy on both known and unknown classes for our method and two baseline methods across various open-set tasks. The results are shown in Figure 6. As illustrated, while the accuracy on unknown classes is comparable across methods, our ETA consistently achieves higher accuracy on known classes. In contrast, baseline methods often suffer a sharp performance drop on known classes, indicating their inability to learn balanced node representations (which also explains their lower HS scores). In comparison, our ETA maintains a

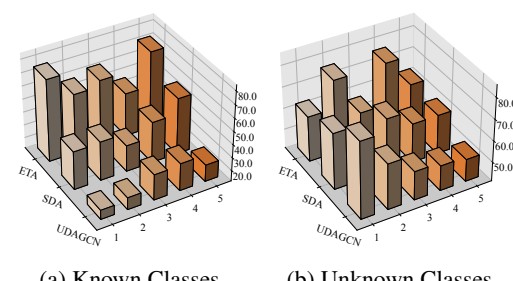

(a) Known Classes     (b) Unknown Classes

Figure 6: Differentiation results between known and unknown classes in various experiments.

more balanced accuracy between known and unknown classes, highlighting its effectiveness in learning more generalizable class representations under open-set scenarios.

### 3.6 QUANTIFICATION OF UNCERTAINTY

To demonstrate that uncertainty can serve as an reliable indicator to distinguish unknown nodes from known nodes, we record the corresponding uncertainty values of source-domain known nodes $S_k$, target-domain known nodes $T_k$, and target-domain unknown nodes $T_u$ over 10 OS-GDA experiments. The results are shown in Figure 7. We can clearly observe that the uncertainty of known nodes in both the source and target domains is generally low, while the uncertainty of unknown nodes in the target domain is significantly higher. This indicates that node uncertainty effectively captures the inherent prop-

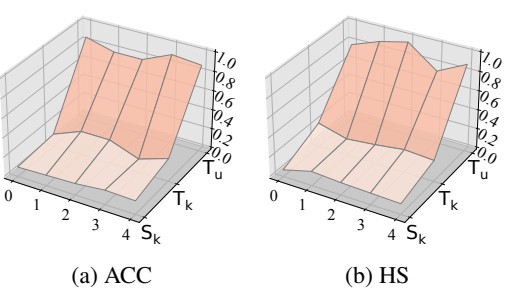

(a) ACC          (b) HS

Figure 7: Impact of uncertainty across node types.

erties of known and unknown classes of nodes in OSGDA scenarios, making it a reliable indicator for distinguishing unknown class nodes. further aligning with our theoretical insights in Section 2.3.

## 4 CONCLUSION

This paper addresses the challenging open-set graph domain adaptation problem, where the target domain contains previously unseen classes. We propose a novel dual evidence-aware uncertainty learning framework ETA, which first adopts a dual-branch encoder to capture both local structures and global semantics, while integrating evidential deep learning to quantify uncertainty through Dirichlet distributions. Based on this uncertainty, our ETA detects unknown target nodes and builds cross-domain neighborhoods via latent MixUp, creating more informative and transferable virtual samples. We also introduce evidential adjacency-consistent uncertainty to measure neighborhood consistency, providing auxiliary supervision for robust domain alignment. Extensive experiments demonstrate that our proposed ETA consistently outperforms existing approaches. In future work, we plan to extend ETA to more challenging learning paradigms, such as label-noised learning and class-imbalanced learning, and further explore its applicability in inherent real-world OSGDA scenarios such as dynamic graph systems (e.g., cybersecurity intrusion detection) or under multi-modal scenarios.

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

# A DEFINITION

## A.1 DIRICHLET DISTRIBUTION

**Definition 1.** (*Dirichlet distribution*) *The Dirichlet distribution is a multivariate probability distribution parameterized by a vector $\boldsymbol{\alpha} = [\alpha_1, \ldots, \alpha_K]$, where each $\alpha_k > 0$. The probability density function (PDF) of the Dirichlet distribution is defined as:*

$$D(\boldsymbol{p}|\boldsymbol{\alpha}) = \begin{cases} \frac{1}{B(\boldsymbol{\alpha})} \prod_{k=1}^{K} p_k^{\alpha_k - 1} & \text{for } \boldsymbol{p} \in \mathcal{S}_K, \\ 0 & \text{otherwise}, \end{cases} \tag{16}$$

*where $\mathcal{S}_K$ denote the $K$-dimensional unit simplex, which can defined as*

$$\mathcal{S}_K = \left\{ \mathbf{p} \mid \sum_{i=1}^{K} p_i = 1 \text{ and } 0 \leq p_1, \ldots, p_K \leq 1 \right\}, \tag{17}$$

*and $B(\boldsymbol{\alpha})$ denotes the $K$-dimensional multinomial beta function. The Dirichlet distribution can be considered as the conjugate prior of the multinomial distribution.*

## A.2 DEMPSTER'S COMBINATION RULE

**Definition 2.** (*Dempster's Combination Rule*) *Given the belief and the uncertainty mass assignments from two sets $\mathcal{M}_{v,edge} = \left\{ \{b_{v,edge}^k\}_{k=1}^K, u_{v,edge} \right\}$ and $\mathcal{M}_{v,path} = \left\{ \{b_{v,path}^k\}_{k=1}^K, u_{v,path} \right\}$, the joint mass $\mathcal{M}_v = \left\{ \{b_v^k\}_{k=1}^K, u_v \right\}$ can be calculated as:*

$$\mathcal{M}_v = \mathcal{M}_{v,edge} \oplus \mathcal{M}_{v,path}. \tag{18}$$

*The calculation rule can be formulated as:*

$$b_v^k = \frac{1}{1-C}(b_{v,edge}^k b_{v,path}^k + b_{v,edge}^k u_{v,path} + b_{v,path}^k u_{v,edge}), u_v = \frac{1}{1-C} u_{v,edge} u_{v,path}, \tag{19}$$

*where $C = \sum_{i \neq j} b_{v,edge}^i b_{v,path}^j$ quantifies the conflict between the two mass sets. The normalization is achieved by applying a scaling factor $\frac{1}{1-C}$.*

# B PROOF OF THEOREM

## B.1 PROOF OF THEOREM 1

*Proof.* Assume the following:

(A1) For each known class $c \in \mathcal{C}_s$, there exists a prototype $m_c \in Z$.

(A2) For each class $c$ and each evidence dimension $k$, there exists a nonnegative, nonincreasing function $\varphi_{c,k} : [0, \infty) \to [0, \infty)$ such that

$$e_z^k \leq \varphi_{c,k}(\|z - m_c\|), \quad \text{whenever } c = \arg\min_{c'} \|z - m_{c'}\|.$$

For each node $v$, let $c^\star = \arg\min_c \|Z_v - m_c\|$, with radius $r = \|Z_v - m_{c^\star}\|$. By assumption (A2), for each evidence $k$ we have

$$e_v^k \leq \varphi_{c^\star, k}(r). \tag{20}$$

Summing over $k$ evidence gives

$$\sum_{k=1}^{K} e_v^k \leq \sum_{k=1}^{K} \varphi_{c^\star, k}(r) \leq \max_c \sum_{k=1}^{K} \varphi_{c,k}(r). \tag{21}$$

Hence

$$S_v = \sum_{k=1}^{K} e_v^k + K \leq K + \max_c \sum_{k=1}^{K} \varphi_{c,k}(r) = G(r). \tag{22}$$

If $\min\limits_c \|z - m_c\| \geq d$, then $r \geq d$, and since each $\varphi_{c,k}$ is nonincreasing, $G(r) \leq G(d)$. Thus $S_v \leq G(d)$, which implies

$$u_v = \frac{K}{S_v} \geq \frac{K}{G(d)}. \tag{23}$$

$\square$

**Remark 1.** *This theorem provides a formal link between embedding space geometry and evidential uncertainty. It shows that if a target node $v$ lies far away from all source class prototypes, then its total evidence $S_v$ is provably bounded above, leading to a provable lower bound on uncertainty $u_v$. This theoretical guarantee supports the use of uncertainty threshold for node detection in ETA.*

**Corollary 1** (**Quadratic decay under local curvature**). *Suppose further that the evidence projection head $\mathbf{h}$ is twice differentiable near each prototype $m_c$, and that for every component $h^k$ the Hessian at $m_c$ satisfies*

$$\nabla^2 h^k(m_c) \preceq -\mu_k I, \mu \succ \mathbf{0}. \tag{24}$$

*Then there exist $r_0 > 0$ and $C > 0$ such that for all $0 \leq r \leq r_0$,*

$$\sum_{k=1}^{K} e_{m_\delta}^k \leq \sum_{k=1}^{K} e_{m_c}^k - Cr^2, \forall \|\delta\| = r. \tag{25}$$

*Where $m_\delta := m_c + \delta$, denotes a node that near the prototype $m_c$. Consequently, for any node $v$ with $\min\limits_c \|Z_v - m_c\| = d \leq r_0$ we have*

$$u_v \geq \frac{K}{\max\limits_c (S_{m_c} - Cd^2)}. \tag{26}$$

*Proof.* Fix a component $k$ and a vector $\delta$ with $\|\delta\| = r \leq r_0$. By Taylor's theorem with the Hessian evaluated at an intermediate point, there exists $\theta \in (0, 1)$ such that

$$h^k(m_c + \delta) = h^k(m_c) + \nabla h^k(m_c)^\top \delta + \tfrac{1}{2}\delta^\top \nabla^2 h^k(m_c + \theta\delta)\,\delta. \tag{27}$$

By assumption $\nabla h^k(m_c) = 0$, hence the linear term disappears. Using the Hessian upper bound in the ball $\|\delta\| \leq r_0$, we get

$$\delta^\top \nabla^2 h^k(m_c + \theta\delta)\,\delta \leq -\frac{\mu_k}{2}\|\delta\|^2. \tag{28}$$

By substituting the former in the equation 28, we can get

$$h^k(m_c + \delta) \leq h^k(m_c) - \frac{\mu_k}{4}\|\delta\|^2. \tag{29}$$

Summing the above inequality over all evidence $k = 1, \dots, K$ gives

$$\sum_{k=1}^{K} e_{m_\delta}^k \leq \sum_{k=1}^{K} e_{m_c}^k - \frac{1}{4}\left(\sum_{k=1}^{K} \mu_k\right) r^2, \tag{30}$$

where $m_\delta := m_c + \delta$. Set $C = \frac{1}{4}\sum_{k=1}^{K} \mu_k > 0$ to conclude the stated bound, and we can get

$$S_{m_\delta} = \sum_{k=1}^{K} e_{m_\delta}^k + K \leq \sum_{k=1}^{K} e_{m_c}^k + K - Cr^2 \leq \max_c(S_{m_c} - Cd^2). \tag{31}$$

Since $u_v = \frac{K}{S_v}$ and for $\max\limits_c(S_{m_c} - Cd^2)$, we can get the final result as

$$u_v = \frac{K}{S_v} \geq \frac{K}{\max\limits_c(S_{m_c} - Cd^2)}. \tag{32}$$

$\square$

**Remark 2.** *The corollary shows that for a neighborhood $v$ of a prototype $m_c$, the total evidence $S_v$ decays at least quadratically in the distance $r = \|Z_v - m_c\|$. Since uncertainty is defined as $u_v = K/S_v$, this quadratic decay in $S_v$ implies a quantitative increase in uncertainty:*

$$u_v \geq \frac{K}{S_{m_c} - Cr^2}, \quad \text{for all } r \text{ such that } S_{m_c} - Cr^2 > 0.$$

*This bound has three important implications. First, it formalizes the intuition that the nodes farther away from known prototypes must be assigned larger epistemic uncertainty. Second, the rate of increase is explicit: uncertainty grows at least inversely with a quadratic function of distance, providing a concrete mechanism to separate known and unknown classes in OSGDA settings. Third, the condition $S_{m_c} - Cr^2 > 0$ restricts the guarantee to a finite neighborhood, beyond this region, the theoretical bound may become vacuous, though empirically the same monotonic trend often persists.*

## B.2 PROOF OF PROPOSITIONS

**Proposition 1.** *For any $\alpha_v^k \geq 1$, the inequality $(\mathcal{L}_v^k)^{var} < (\mathcal{L}_v^k)^{err}$ satisfied.*

*Proof.* When $y_v^k = 0$, then $(\mathcal{L}_v^k)^{err} = \frac{(\alpha_v^k)^2}{S_v^2}$. As $\frac{(S_v - \alpha_v^k)}{(S_v+1)} < 1$ and $\frac{\alpha_v^k}{S_v^2} \leq \frac{(\alpha_v^k)^2}{S_v^2}$ we obtain

$$\frac{\alpha_v^k(S_v - \alpha_v^k)}{S_v^2(S_v + 1)} < \frac{(\alpha_v^k)^2}{S_v^2}. \tag{33}$$

Now consider the case $y_v^k = 1$. Then

$$(\mathcal{L}_v^k)^{err} = \left(1 - \frac{\alpha_v^k}{S_v}\right)^2 = \frac{(S_v - \alpha_v^k)^2}{S_v^2}. \tag{34}$$

As $(S_v - \alpha_v^k) > \frac{\alpha_v^k}{S_v+1}$, we attain

$$\frac{\alpha_v^k(S_v - \alpha_v^k)}{S_v^2(S_v + 1)} < \frac{(S_v - \alpha_v^k)^2}{S_v^2}. \tag{35}$$

Thus in both cases $(\mathcal{L}_v^k)^{var} < (\mathcal{L}_v^k)^{err}$. $\square$

**Proposition 2.** *For a given sample $v$ with the correct label $k$, $L_v^{err}$ decreaces when new evidence is added to $\alpha_v^k$ and increases when evidence is removed from $\alpha_v^k$*

*Proof.* Let $\delta$ denote additional evidence added to the Dirichlet parameter $\alpha_v^k$. Then $L_v^{err}$ is updated as

$$\hat{L}_v^{err} = \left(1 - \frac{\alpha_v^k + \delta}{S_v + \delta}\right)^2 + \sum_{l \neq k}\left(\frac{\alpha_v^l}{S_v + \delta}\right)^2. \tag{36}$$

For $\delta > 0$ we have

$$\left(1 - \frac{\alpha_v^k + \delta}{S_v + \delta}\right)^2 < \left(1 - \frac{\alpha_v^k}{S_v}\right)^2 \tag{37}$$

and

$$\sum_{l \neq k}\left(\frac{\alpha_v^l}{S_v + \delta}\right)^2 < \sum_{l \neq k}\left(\frac{\alpha_v^l}{S_v}\right)^2. \tag{38}$$

Hence $\hat{L}_v^{err} < L_v^{err}$. Similarly, for $\delta < 0$ the inequalities reverse, so $\hat{L}_v^{err} > L_v^{err}$. $\square$

**Proposition 3.** *For a given sample $v$ with the correct class label $j$, $L_v^{err}$ decreases when some evidence is removed from the biggest Dirichlet parameter $\alpha_v^l$ such that $l \neq j$.*

*Proof.* Let the expected value of the predicted Dirichlet distribution for sample $v$ be $\hat{p}_v = (\hat{p}_v^1, \ldots, \hat{p}_v^k)$. When some evidence is removed from $\alpha_v^l$, $\hat{p}_v^l$ decreases by $\delta_v^l > 0$. As a result, $\hat{p}_v^k$ for all $k \neq l$ increases by $\delta_v^k > 0$ with $\sum_{k \neq l} \delta_v^k = \delta_v^l$, since the expectations must sum to one. Let $\tilde{p}_v^l$ denote the updated expected value for the $l^{th}$ component of the Dirichlet distribution after removal of evidence. Then, before the removal $L_v^{err}$ can be written as

$$L_v^{err} = (1 - \hat{p}_v^j)^2 + \left( \tilde{p}_v^l + \sum_{k \neq l} \delta_v^k \right)^2 + \sum_{k \notin \{j,l\}} (\hat{p}_v^k)^2. \tag{39}$$

After the removal of evidence, it is updated as

$$\tilde{L}_v^{err} = (1 - \hat{p}_v^j - \delta_v^j)^2 + (\tilde{p}_v^j)^2 + \sum_{k \notin \{j,l\}} (\hat{p}_v^k + \delta_v^k)^2. \tag{40}$$

The difference of $L_v^{err} - \tilde{L}_v^{err}$ is

$$L_v^{err} - \tilde{L}_v^{err} = \underbrace{2(1 - \hat{p}_v^j)\delta_v^j}_{\geq 0} + 2 \left( \tilde{p}_v^l \sum_{k \neq l} \delta_v^k - \sum_{k \notin \{j,l\}} \hat{p}_v^k \delta_v^k \right) + \underbrace{\left( \left( \sum_{k \neq l} \delta_v^k \right)^2 - \sum_{k \neq l} (\delta_v^k)^2 \right)}_{\geq 0}, \tag{41}$$

which is always positive provided $\hat{p}_v^l > \tilde{p}_v^l \geq \hat{p}_v^k$ for $k \neq j$, and is maximized as $\hat{p}_v^l$ increases. □

## C SUPPLEMENTAL EXPERIMENT RESULTS AND ANALYSIS

Table 1: Details of various experiments with different unknown classes (ACC (%) and HS (%) ).

| Removed labels | A⇒D | | D⇒A | | A⇒C | | C⇒A | | C⇒D | | D⇒C | |
|---|---|---|---|---|---|---|---|---|---|---|---|---|
| | ACC | HS | ACC | HS | ACC | HS | ACC | HS | ACC | HS | ACC | HS |
| $\{0,1\}$ | 68.65 | 66.34 | 52.06 | 51.94 | 69.33 | 68.91 | 60.01 | 59.13 | 73.03 | 71.39 | 62.07 | 61.71 |
| $\{0,2\}$ | 60.34 | 59.49 | 58.28 | 58.56 | 66.96 | 66.20 | 59.09 | 56.93 | 58.33 | 51.32 | 68.05 | 68.03 |
| $\{0,3\}$ | 62.01 | 60.74 | 47.53 | 49.37 | 60.97 | 59.82 | 64.21 | 59.01 | 56.38 | 59.38 | 57.75 | 59.43 |
| $\{0,4\}$ | 55.00 | 55.94 | 47.38 | 47.32 | 59.69 | 57.86 | 65.32 | 58.74 | 58.08 | 58.85 | 54.24 | 52.46 |
| $\{1,2\}$ | 62.98 | 63.23 | 70.89 | 68.74 | 67.73 | 67.23 | 69.65 | 62.63 | 70.64 | 68.45 | 70.89 | 68.74 |
| $\{1,3\}$ | 57.37 | 57.59 | 53.69 | 54.54 | 58.93 | 59.72 | 60.74 | 45.70 | 68.91 | 68.02 | 62.66 | 57.04 |
| $\{1,4\}$ | 60.08 | 61.80 | 51.76 | 44.74 | 60.97 | 60.80 | 59.63 | 58.88 | 70.06 | 69.66 | 57.28 | 49.65 |
| $\{2,3\}$ | 60.31 | 59.23 | 60.68 | 61.26 | 61.47 | 63.37 | 60.22 | 62.05 | 68.34 | 62.21 | 59.27 | 53.33 |
| $\{2,4\}$ | 64.57 | 61.84 | 53.03 | 51.68 | 56.44 | 56.86 | 58.33 | 57.56 | 57.75 | 58.42 | 53.11 | 31.35 |
| $\{3,4\}$ | 67.34 | 63.42 | 46.78 | 49.43 | 65.11 | 64.96 | 61.07 | 61.11 | 66.31 | 67.32 | 54.27 | 57.30 |
| Average | 61.87 | 60.96 | 54.21 | 53.76 | 62.76 | 62.57 | 61.83 | 58.17 | 64.78 | 63.50 | 59.96 | 55.90 |

Table 2: Detailed ACC (%) of known classes and unknown classes of OSGDA experiments.

| Removed labels | A⇒D | | D⇒A | | A⇒C | | C⇒A | | C⇒D | | D⇒C | |
|---|---|---|---|---|---|---|---|---|---|---|---|---|
| | $ACC_k$ | $ACC_u$ | $ACC_k$ | $ACC_u$ | $ACC_k$ | $ACC_u$ | $ACC_k$ | $ACC_u$ | $ACC_k$ | $ACC_u$ | $ACC_k$ | $ACC_u$ |
| $\{0,1\}$ | 58.00 | 77.50 | 54.61 | 49.52 | 75.93 | 63.08 | 67.29 | 52.47 | 64.23 | 80.34 | 58.00 | 65.93 |
| $\{0,2\}$ | 73.37 | 50.03 | 54.41 | 63.41 | 58.54 | 76.18 | 65.84 | 50.15 | 74.31 | 39.20 | 68.34 | 67.72 |
| $\{0,3\}$ | 60.30 | 61.18 | 43.91 | 56.38 | 62.93 | 57.01 | 53.62 | 65.61 | 49.85 | 73.42 | 52.28 | 68.83 |
| $\{0,4\}$ | 49.70 | 63.97 | 38.82 | 60.60 | 46.50 | 76.57 | 50.31 | 70.57 | 49.41 | 72.75 | 41.98 | 69.93 |
| $\{1,2\}$ | 65.85 | 60.80 | 59.51 | 81.34 | 72.46 | 62.70 | 48.20 | 89.36 | 61.00 | 77.98 | 59.51 | 81.34 |
| $\{1,3\}$ | 50.18 | 68.60 | 48.00 | 62.91 | 56.99 | 62.73 | 78.29 | 32.27 | 71.50 | 64.86 | 70.08 | 48.09 |
| $\{1,4\}$ | 69.97 | 55.34 | 69.35 | 33.02 | 53.79 | 69.92 | 52.16 | 67.58 | 63.78 | 76.74 | 73.08 | 37.60 |
| $\{2,3\}$ | 70.25 | 51.20 | 59.66 | 62.96 | 57.59 | 70.45 | 56.03 | 69.52 | 59.12 | 65.64 | 65.80 | 44.83 |
| $\{2,4\}$ | 55.24 | 70.23 | 57.20 | 47.13 | 53.57 | 60.57 | 46.86 | 74.57 | 53.16 | 64.84 | 76.27 | 19.73 |
| $\{3,4\}$ | 74.05 | 55.46 | 40.53 | 63.34 | 60.58 | 70.03 | 62.93 | 59.41 | 83.54 | 56.38 | 49.11 | 68.76 |

### C.1 DETAILED OSGDA EXPERIMENTS RESULTS

Our open-set graph domain adaptation (OSGDA) experiments are conducted on three benchmark datasets (ACMv9, Citationv1, DBLPv7). In each experiment, one dataset is selected as the source

domain, while the other two serve as target domains. To simulate the open-set scenario, we remove two out of five classes in turn. We present the detailed results of ETA on the OSGDA task here, including average class accuracy on known classes (ACC$_k$), accuracy on unknown classes (ACC$_u$), average per-class accuracy over the entire domain (ACC), and the H-score (HS), as shown in Tables 1 and Table 2. We also provide visualization analyses in Figure 1. From the figures, we can observe some performance variations under different open-set settings (i.e., different combinations of source and target domains and unknown categories). In certain cases, the model scores are relatively lower, indicating that domain adaptation becomes particularly challenging under specific open-set scenarios. Nevertheless, our method maintains stable performance across different settings — for example, ACC consistently remains within the range [50, 70] — demonstrating the effectiveness and robustness of our approach in various open-set domain adaptation situations.

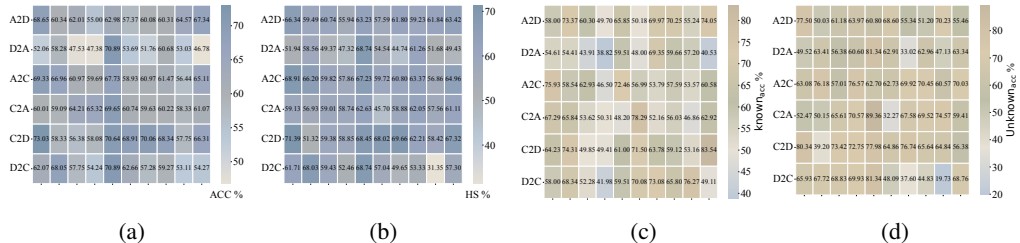

| (a) | (b) | (c) | (d) |

Figure 1: ACC (%), HS (%) of OSGDA experiments ( (a) and (b) ) and ACC (%) of known and unknown classes of OSGDA experiments ( (c) and (d) ). Every column represents various experiment with different unknown classes while the vertical axis represents different source and target pairs.

## C.2 ABLATION STUDY

Table 3: The results of ablation study experiments (ACC (%) and HS (%) ).

| Methods | A⇒D | | D⇒A | | A⇒C | | C⇒A | | C⇒D | | D⇒C | |
|---|---|---|---|---|---|---|---|---|---|---|---|---|
| | ACC | HS | ACC | HS | ACC | HS | ACC | HS | ACC | HS | ACC | HS |
| w/o $\mathcal{L}_{da}$ | 55.54 | 54.47 | 44.65 | 43.31 | 52.34 | 50.06 | 48.82 | 46.68 | 55.58 | 54.42 | 50.06 | 48.84 |
| w/o $\mathcal{L}_{evi}$ | 45.54 | 42.21 | 40.01 | 38.85 | 44.43 | 42.31 | 38.84 | 34.41 | 46.68 | 40.06 | 45.53 | 44.48 |
| w/o PAN | 59.86 | 57.40 | 47.18 | 46.51 | 62.47 | 56.31 | 52.54 | 50.01 | 65.58 | 62.94 | 56.65 | 54.49 |
| w/o GCN | 60.33 | 55.34 | 45.42 | 42.23 | 60.03 | 54.46 | 50.51 | 46.65 | 63.31 | 58.84 | 54.49 | 52.28 |
| ETA$_{entropy}$ | 60.32 | 56.63 | 50.83 | 47.66 | 60.33 | 57.76 | 55.54 | 52.28 | 63.38 | 60.28 | 56.64 | 53.36 |
| ETA | **68.65** | **66.34** | **52.06** | **51.94** | **69.33** | **68.91** | **60.01** | **59.13** | **73.03** | **71.39** | **62.07** | **61.71** |

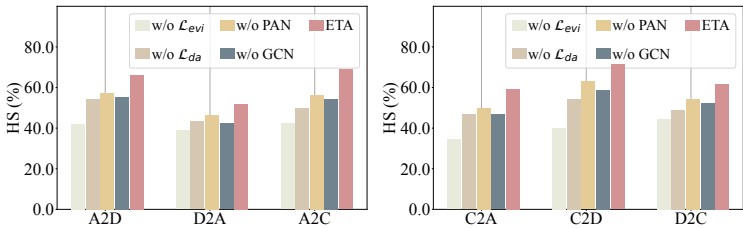

Figure 2: Visualization of ablation study on six data pairs (HS (%) ).

To investigate the effectiveness of each component of ETA, we conducted ablation studies, with detailed results presented in Table 3 and a corresponding visualization analysis shown in Figure 2. As illustrated in the Table 3, modifying or removing any part of our method results in a noticeable performance drop in both ACC and HS. The most significant degradation occurs when the evidence loss is removed, highlighting the importance of leveraging evidential values to distinguish unknown-class nodes and prevent them from negatively affecting the training of known-class nodes. Figure 2 shows the HS visualization from the ablation study, clearly demonstrating that the trend of HS aligns

with that of ACC. This consistency further validates the effectiveness of each component in ETA. In addition, to quantify the benifit of evidential deep learning in OSGDA, we conduct an ablation study comparing ordinary predictive entropy (ETA$_{entropy}$) with uncertainty mass (ETA). The results clearly show that replacing the uncertainty mass with entropy leads to noticeable performance degradation, further validating the superiority of evidence-based uncertainty modeling in OSGDA.

## C.3 SENSITIVITY ANALYSIS

To investigate the impact of various hyperparameter values on our method, we conducted a series of hyperparameter sensitivity experiments.

Table 4: Sensitivity analysis of different $k$ (ACC (%) and HS (%) ).

| $k$ | A⇒D | | D⇒A | | A⇒C | | C⇒A | | C⇒D | | D⇒C | |
|---|---|---|---|---|---|---|---|---|---|---|---|---|
| | ACC | HS | ACC | HS | ACC | HS | ACC | HS | ACC | HS | ACC | HS |
| 1 | 65.33 | 61.74 | 48.18 | 47.86 | 66.34 | 62.30 | 56.32 | 55.98 | 68.89 | 66.32 | 55.34 | 54.80 |
| 3 | **68.65** | **66.34** | **52.06** | **51.94** | **69.33** | **68.91** | **60.01** | **59.13** | **73.03** | **71.39** | **62.07** | **61.71** |
| 5 | 66.37 | 62.38 | 49.34 | 48.52 | 67.90 | 64.48 | 58.83 | 57.46 | 68.04 | 67.76 | 57.79 | 55.80 |

**Impact of** $k$: Table 4 shows the model's performance with different values of $k$ (i.e., $k = \{1, 3, 5\}$). As observed from both the table, the model achieves the best performance when $k = 3$. As discussed in 3.3, setting $k$ too small leads to insufficient neighbor information, while setting it too large increases the likelihood of introducing noisy, inconsistent neighbors. Choosing $k = 3$ strikes a balance between information sufficiency and noise control, thereby facilitating more effective domain alignment.

Table 5: Sensitivity analysis of different uncertainty threshold $\eta$ (ACC (%) and HS (%)).

| $\eta$ | A⇒D | | D⇒A | | A⇒C | | C⇒A | | C⇒D | | D⇒C | |
|---|---|---|---|---|---|---|---|---|---|---|---|---|
| | ACC | HS | ACC | HS | ACC | HS | ACC | HS | ACC | HS | ACC | HS |
| 0.55 | 58.84 | 56.62 | 45.34 | 44.47 | 55.53 | 54.43 | 54.43 | 54.09 | 63.38 | 62.01 | 54.48 | 54.08 |
| 0.60 | 62.48 | 62.01 | 48.84 | 46.65 | 63.34 | 61.09 | 58.84 | 56.65 | 68.21 | 66.74 | 59.91 | 55.56 |
| 0.65 | **68.51** | **65.10** | **55.71** | **54.42** | **73.50** | **71.94** | **60.06** | **59.99** | **71.81** | **68.70** | **57.43** | **55.51** |
| 0.70 | 66.63 | 62.01 | 47.74 | 46.96 | 66.63 | 64.41 | 57.86 | 54.43 | 69.01 | 68.04 | 60.01 | 56.62 |
| 0.75 | 62.34 | 59.89 | 44.43 | 54.42 | 64.43 | 60.81 | 55.58 | 54.69 | 64.45 | 63.31 | 55.18 | 53.29 |

**Impact of** $\eta$: We conducted experiments with $\eta = \{0.55, 0.60, 0.65, 0.70, 0.75\}$, with the results presented in Table 5. The table reveals that extreme values of $\eta$ (either too high or too low) degrade the model's performance. As analyzed in 3.3, an overly small $\eta$ causes many known-class nodes to be mistakenly treated as unknown, while an overly large $\eta$ results in too many unknown-class nodes being classified as known, both introducing noise into the training process. Thus, we select $\eta = 0.65$ to achieve a clearer boundary between known and unknown classes, thereby enhancing model performance on the OSGDA task.

## C.4 PERFORMANCE WITH DIFFERENT BACKBONES

To verify the generalizability of ETA, we conducted experiments using various classic GNNs (GCN (Kipf & Welling, 2017), GAT (Veličković et al., 2018), GraphSAGE (Hamilton et al., 2017), and PAN (Ma et al., 2020)) as the backbone networks for the two branches. The results are shown in Table 6 and Figure 3. As illustrated, for the same OSGDA task, different backbone networks yield comparable performance; similarly, the same backbone combination performs well across different tasks. These results demonstrate that our method consistently achieves strong performance regardless of the underlying GNN architecture, effectively enhancing the ability to capture node information in open-set scenarios, and validating the generalizability of our approach.

## C.5 VISUALIZATION ANALYSIS

To further explore the effects of domain alignment, we present the t-SNE visualization of the nodes' embedding before and after alignment learned by SDA, UAGA and our ETA, and the results are

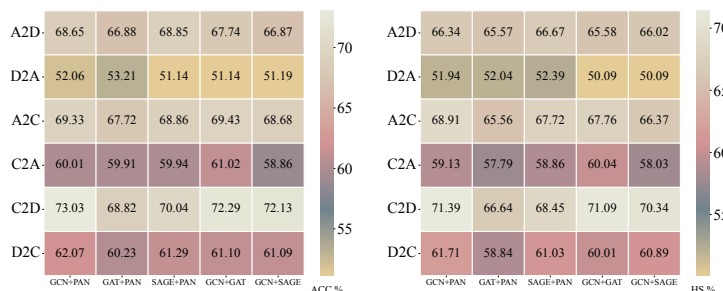

Figure 3: Performance of ETA with different backbones (ACC (%) and HS (%) ).

Table 6: Detailed results of ETA with different backbones (ACC (%) and HS (%) ).

| Backbone | A⇒D | | D⇒A | | A⇒C | | C⇒A | | C⇒D | | D⇒C | |
|---|---|---|---|---|---|---|---|---|---|---|---|---|
| | ACC | HS | ACC | HS | ACC | HS | ACC | HS | ACC | HS | ACC | HS |
| GCN+PAN | 68.65 | 66.34 | 52.06 | 52.94 | 69.33 | 68.91 | 60.01 | 59.13 | 73.03 | 71.39 | 62.07 | 61.71 |
| GAT+PAN | 66.88 | 65.57 | 53.21 | 52.04 | 67.72 | 65.56 | 59.91 | 57.79 | 68.82 | 66.64 | 60.23 | 58.84 |
| GraphSAGE+PAN | 68.85 | 66.67 | 51.14 | 52.39 | 68.86 | 67.72 | 59.94 | 58.86 | 70.04 | 68.45 | 61.29 | 61.03 |
| GCN+GAT | 67.74 | 65.58 | 51.14 | 50.09 | 69.43 | 67.76 | 61.02 | 60.04 | 72.29 | 71.09 | 61.10 | 60.01 |
| GCN+GraphSAGE | 66.87 | 66.02 | 51.19 | 50.09 | 68.68 | 66.37 | 58.86 | 58.03 | 72.13 | 70.34 | 61.09 | 60.89 |

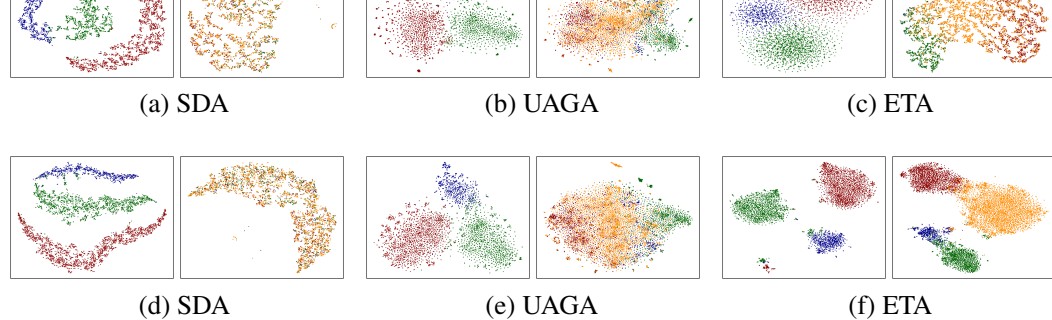

(a) SDA        (b) UAGA        (c) ETA

(d) SDA        (e) UAGA        (f) ETA

Figure 4: T-SNE visualization of source (left) and target (right) domain node embedding before domain alignment (upper) and after domain alignment (lower).

shown in Figure 4. From the t-SNE visualizations, we observe that all three methods are able to learn reasonably good representations on the source domain, both before and after domain alignment. However, the situation differs on the target domain. Prior to alignment, the representations learned by all methods on the target domain appear highly scattered and unstructured. After applying domain alignment, both UAGA and ETA exhibit noticeably clearer cluster boundaries in the target domain, whereas SDA still produces relatively disordered representations. Moreover, across both the source and target domains, our ETA learns the most well-separated class boundaries. This demonstrates that domain alignment module of our framework is more effective at capturing cross-domain invariant knowledge and achieving robust domain-level alignment.

## D   PERFORMANCE ON ADDITIONAL DATASETS

To further validate our ETA, we conduct experiments on three widely used airport datasets (Ribeiro et al., 2017) and two blog datasets (Li et al., 2015), namely USA (U), Brazil (B), Europe (E) for the airport datasets (totally 4 classes) and Blog1 (B1), Blog2 (B2) for the blog datasets (totally 6 classes). For each experiment, we choose two classes as unknown classes, and the others remain known. Besides, we choose some competitive baselines for comparison, the results are provided in Table 7. As shown in the results, our method consistently outperforms the baselines on both the

airport and blog datasets under the OSGDA setting, verifying the generality and generalization ability of our framework. Meanwhile, consistent with our previous experiments, our method demonstrates significant advantages on HS, which proves the robustness and superiority of our approach.

Table 7: Performance on airport and blog datasets.

| Method | U⇒B | | U⇒E | | B⇒E | | B1⇒B2 | |
|---|---|---|---|---|---|---|---|---|
| | ACC | HS | ACC | HS | ACC | HS | ACC | HS |
| Speg-Reg | 33.24 | 22.41 | 28.32 | 22.87 | 27.18 | 13.24 | 29.67 | 18.52 |
| SDA | 49.06 | 40.53 | 42.89 | 33.38 | 45.69 | 35.31 | 52.35 | 43.96 |
| ETA (Ours) | **56.31** | **52.43** | **52.53** | **47.89** | **50.38** | **48.34** | **62.23** | **56.62** |

## E  RELATED WORK

### E.1  GRAPH DOMAIN ADAPTATION

As a branch of graph transfer learning, graph domain adaptation (GDA) enables knowledge to be transferred from a source domain with abundant labels to a target domain with limited labels. This often requires addressing the domain shift problem caused by structural or distributional differences between domains to achieve cross-domain knowledge transfer (Dai et al., 2022; Liu et al., 2023). Recently, GDA methods have generally followed two directions: metric-based and adversarial-based approaches. Metric-based methods reduce the domain discrepancy by minimizing a predefined metric (e.g., Maximum Mean Discrepancy, MMD) to align feature distributions across domains (Gretton et al., 2012; Shen et al., 2020; Wu et al., 2023). In contrast, adversarial-based methods leverage a generator-discriminator training scheme to help GNNs learn domain-invariant representations, thereby enabling effective knowledge transfer (Dai et al., 2022; Qiao et al., 2023). For example, UDAGCN (Wu et al., 2020a) employs dual graph convolution networks and attention mechanisms to transfer knowledge across graphs, optimizing multiple loss functions to achieve domain adaptation in node classification tasks. ASN (Zhang et al., 2021) reduces domain discrepancy by disentangling domain-specific and domain-shared information and combining local and global consistency through adversarial learning. However, most existing GDA techniques assume a shared label space between the source and target domains (i.e., closed-set GDA). In real-world scenarios, this assumption often does not hold, and domains may only partially share labels — a setting known as open-set GDA (OSGDA). To address this, SDA and UAGA (Wang et al., 2024; Shen et al., 2025) was proposed as a method tailored for OSGDA. Despite its effectiveness, SDA and UAGA suffers from performance imbalance between shared and private classes. To overcome this limitation, we propose a novel method inspired by evidence theory, which integrates dual-branch evidence representations to accurately distinguish between shared and private classes. Based on this distinction, we perform domain alignment, which mitigates the class imbalance issue observed in SDA and UAGA. Our approach enables the model to learn more discriminative and balanced semantic representations for different classes, thereby facilitating more effective knowledge transfer in OSGDA scenarios.

### E.2  EVIDENTIAL DEEP LEARNING

Evidence theory, also known as Dempster-Shafer theory (Shafer, 1976), provides a general framework for reasoning under uncertainty. Evidential Deep Learning (EDL) (Sensoy et al., 2018; Malinin & Gales, 2018) introduces this theory into the neural network paradigm. Unlike the traditional softmax output that produces deterministic probabilities, EDL outputs an "amount of evidence" for each class, which parameterizes a Dirichlet distribution, enabling both class prediction and uncertainty estimation. EDL employs a specialized evidential loss that encourages the model to produce high-confidence predictions for correct classifications and high uncertainty for incorrect ones, thereby reducing overconfident misclassifications, which is widely used in the tasks that requiring uncertainty modeling (Sensoy et al., 2020; Shi et al., 2020; Chen et al., 2022). For example, (Bao et al., 2021) use the uncer-tainty obtained by EDL to distinguish between the known and unknown samples for the open set action recognition task; DECL (Qin et al., 2022) integrates a novel cross-modal evidential learning paradigm that captures and models the uncertainty introduced by noise, thereby enhancing the robustness and reliability of cross-modal retrieval. Recently, EAAF (Pei et al., 2024) achieves

fine-grained meta-knowledge aggregation through evidential prediction uncertainty and ensures reliable semantic propagation in the target domain via evidential adjacency-consistent uncertainty, thereby demonstrating strong performance on multi-source unsupervised domain adaptation tasks. Current research has rarely applied evidence theory to GNN tasks. However, graph-structured data inherently involves complex relational uncertainties, and the presence of unknown classes in OSGDA scenarios further amplifies the uncertainty in node classification. This makes the OSGDA scenario particularly well-suited for integration with evidence theory. In this work, we propose a method that incorporates the EDL framework into the OSGDA task. By leveraging evidence theory, we aim to model node classification uncertainty from a new perspective, thereby enhancing model performance in open-set graph domain adaptation settings.

## F   DATASETS

The datasets used in our experiments consist of three components:

- **ACMv9 (A)**: The ACM dataset comprises computer science publications released after 2010, spanning diverse research areas including artificial intelligence, machine learning, data mining, computer networks, software engineering, and other related domains.
- **Citationv1 (C)**: This dataset, derived from the Microsoft Academic Graph (MAG), centers on academic papers published before 2008. It provides rich metadata for each paper, including the title, authors, publication year, and citation links.
- **DBLPv7 (D)**: This dataset is a subset of the DBLP (Digital Bibliography & Library Project), concentrating on computer science publications from 2004 to 2008. As one of the most comprehensive and widely utilized bibliographic resources in the field, DBLP indexes academic papers, authors, conferences, and journals across various computer science domains.

Each dataset represents a citation network (Tang et al., 2008), where nodes correspond to academic papers and edges indicate citation relationships between them. Node features are derived from sparse bag-of-words vectors extracted from paper titles, while node labels denote the research domain to which each paper belongs. All datasets share a common label space, each containing five categories, though they represent citation networks from different time periods, resulting in label distribution shifts across domains. To simulate the open-set scenario, we systematically remove two out of the five labels in each experiment to serve as unknown classes, reflecting various open-set conditions. Detailed statistics of the datasets are provided in Table 8.

Table 8: Statistics of citation networks.

| Dataset | Nodes | Edges | Attributes | Label Proportion (%) |
|---------|-------|-------|------------|----------------------|
| ACMv9 | 9,360 | 15,602 | 5,571 | 20.5/29.6/22.5/8.6/18.8 |
| Citationv1 | 8,935 | 15,113 | 5,379 | 25.3/26.0/22.5/7.7/18.5 |
| DBLPv7 | 5,484 | 8,130 | 4,412 | 21.7/33.0/23.8/6.0/15.5 |

## G   BASELINES AND EVALUATION CRITERIA

To validate the effectiveness of our method on the OSGDA task, we selected five categories of baseline models for comparison in our experiments: 1) *Classical GNNs*: GCN (Kipf & Welling, 2017) and GraphSAGE (Hamilton et al., 2017); 2) *Unsupervised domain adaptation (UDA) methods*: DANN (Ganin et al., 2016) and CDAN (Long et al., 2018); 3) *Open-set domain adaptation (OSDA) methods*: OSBP (Saito et al., 2018) and DANCE (Saito et al., 2020); 4) *Closed-set graph domain adaptation (CSGDA) methods*: UDAGCN (Wu et al., 2020a) and ASN (Zhang et al., 2021); 5) *OSGDA method*: SDA (Wang et al., 2024). Besides several criteria that are widely used in classification tasks such as ACC, we also select one criterion that is suitable for the open-set tasks called H-score (HS) (Fu et al., 2020). More details of baselines and HS are shown as follows:

1) The H-score can be calculated as:

$$HS = \frac{2 \times ACC_k \times ACC_u}{ACC_k + ACC_u} \tag{42}$$

2) More information of baselines:

- **Classical GNNs**: GCN (Kipf & Welling, 2017) propagates information across neighbors, while GraphSAGE (Hamilton et al., 2017) generates embeddings inductively by aggregating local neighborhood features for unseen nodes, enabling effective node representation learning.

- **DANN** (Ganin et al., 2016): This method leverages a gradient reversal layer in neural networks to learn features that are discriminative for the source domain but invariant to domain shifts, enabling successful adaptation to target domains with unlabeled data

- **CDAN** (Long et al., 2018): A domain adaptation method that uses conditional adversarial learning, incorporating multilinear and entropy conditioning to improve discriminability and transferability.

- **OSBP** (Saito et al., 2018): An open-set domain adaptation method that uses adversarial training to separate unknown target samples from known ones.

- **DANCE** (Saito et al., 2020): It is a domain adaptation method that handles arbitrary category shifts by combining self-supervised neighborhood clustering and entropy-based feature alignment.

- **UDAGCN** (Wu et al., 2020a): An unsupervised domain adaptive graph convolutional network that leverages dual graph convolution and an attention mechanism to enable knowledge transfer between graphs, optimizing multiple loss functions for graph domain adaptation tasks.

- **ASN** (Zhang et al., 2021): It is a novel model for cross-network node classification that separates domain-private and domain-shared information, combining local and global consistency while using adversarial domain adaptation to reduce distribution discrepancy across networks.

- **SDA** (Wang et al., 2024): It is a novel approach for open-set domain adaptive node classification, which efficiently transfers knowledge from a labeled source graph to an unlabeled target graph, enabling both classification of known nodes and detection of unknown classes in the target domain.

- **UAGA** (Shen et al., 2025): It tackles the OSGDA problem using an unknown-excluded adversarial graph domain alignment approach, selectively aligning target nodes of known classes with the source domain while pushing target nodes of unknown classes away via an adaptation coefficient.

