# OpenReview forum: "ETA: Dual Evidence-Aware Uncertainty Learning for Open-Set Graph Domain Adaptation"
_ICLR.cc/2026/Conference — ICLR 2026 Conference Withdrawn Submission_

### Official Review · Reviewer_xDtZ · 2025-10-27

**Soundness:** 4
**Presentation:** 4
**Contribution:** 3
**Rating:** 6
**Confidence:** 3

**Summary:**

The paper "ETA: Dual Evidence-Aware Uncertainty Learning for Open-Set Graph Domain Adaptation" proposes a novel framework to address the challenging problem of Graph Domain Adaptation (GDA) where the target graph contains new, unknown classes not seen in the source domain.

**Strengths:**

1. The proposal of a dual-branch encoder is well-motivated. Graphs exhibit both local neighborhood effects and long-range dependencies, and capturing evidence from both perspectives before fusing it via the evidential learning framework likely leads to more robust and accurate evidence accumulation for final classification and uncertainty estimation.

2. The introduction of MixUp-based virtual sample generation in the latent space, conditioned on identified unknown nodes and cross-domain neighborhoods, is a clever technique. This MixUp strategy creates an auxiliary supervision signal that encourages smoother transitions between known classes and the unknown class, promoting more robust domain alignment while mitigating the negative transfer caused by the unknown samples.

**Weaknesses:**

1. The performance of EDL-based models can be sensitive to the design of the evidence loss and regularization terms (e.g., the $\mathcal{L}_{\text{evidence}}$ term mentioned in similar literature). The description of the evidential adjacency-consistent uncertainty term suggests a careful loss formulation, but the precise formulation and parameter tuning should be detailed and discussed.

2. While fusing local and global evidence is intuitive, the paper may need clearly explain how the evidence from the two branches (Dirichlet distributions) is formally combined/integrated (e.g., using Dempster-Shafer theory's combination rule or a simpler summation/concatenation of concentration parameters) and why that specific fusion method is optimal for this graph problem.

**Questions:**

1. May provide detailed visualizations (e.g., t-SNE) comparing the latent space representations of ETA versus leading baselines (like SDA and UAGA), explicitly showing the separation of source, target-known, and target-unknown features before and after alignment.

2. Ablation studies may include a variant of ETA that uses a standard confidence-based threshold (like maximum softmax probability or entropy) instead of the full EDL uncertainty for unknown detection, to quantify the specific benefit of EDL.

---

> ### Author Response · Authors · 2025-11-21
>
> Thanks for your insightful comments and valuable feedback. We'll spare no effort to address your concerns in the following.
>
> Q1: May provide detailed visualizations (e.g., t-SNE) comparing the latent space representations of ETA versus leading baselines (like SDA and UAGA), explicitly showing the separation of source, target-known, and target-unknown features before and after alignment.
>
> R1: Thanks for the comment! We present the t-SNE visualizations of the source graph and target graph before and after alignment in our updated manuscript, **please refer to the blue-highlighted content in Appendix C.5 in the revised manuscript.**
>
> Q2: Ablation studies may include a variant of ETA that uses a standard confidence-based threshold (like maximum softmax probability or entropy) instead of the full EDL uncertainty for unknown detection, to quantify the specific benefit of EDL.
>
> R2: Thank you for the question. On six dataset pairs, we treat label 0/1 as the unknown class and compare two criteria for identifying unknown nodes: entropy of the predictive distribution ($\text{ETA}_{entropy}$) and the uncertainty mass produced by EDL ($\text{ETA}$). Both variants adopt the same dynamic thresholding strategy with an initial threshold of 0.6. The results are reported below:
>
> |Method|A2D|D2A|A3C|C2A|C2D|D2C|
> |-|-|-|-|-|-|-|
> $\text{ETA}_{entropy}$|60.32/56.63|50.83/47.66|60.33/57.76|55.54/52.28|63.38/60.28|56.64/53.36|
> $\text{ETA(Ours)}$|68.65/66.34|52.06/51.94|69.33/68.91|60.01/59.13|73.03/71.39|62.07/61.71|
>
> The values in each cell indicate $ACC$/$HS$. As shown in the table, replacing the uncertainty mass with entropy leads to a noticeable performance drop across all dataset pairs, demonstrating the superiority of EDL-based uncertainty mass for the OSGDA task.
>
> We further analyze the reason behind this gap: The maximum value of predictive entropy depends on the total number of classes. In different OSGDA scenarios, the number of known classes varies substantially, which causes large fluctuations in the entropy range. Such instability makes entropy-based thresholding unreliable for distinguishing unknown classes under varying label spaces. **In contrast, the uncertainty mass from EDL has a fixed upper bound of 1 regardless of the number of classes or the specific domain adaptation setting. This inherent stability allows the model to maintain consistent performance across different OSGDA scenarios.**
>
> We have included the ablation study in our updated manuscript, **please refer to the blue-highlighted content in Appendix C.2 in the revised manuscript.**

---

> ### Author Response · Authors · 2025-11-21
>
> Q3: The performance of EDL-based models can be sensitive to the design of the evidence loss and regularization terms (e.g., the $\mathcal{L}\_{evidence}$ term mentioned in similar literature). The description of the evidential adjacency-consistent uncertainty term suggests a careful loss formulation, but the precise formulation and parameter tuning should be detailed and discussed.
>
> R3: Thank you for the insightful comment. In our ETA, two losses are directly associated with the evidence: the **evidence-aware loss** ($\mathcal{L}\_{evi}$), which is responsible for forming class-specific opinions, and the **evidence consistency loss** ($\mathcal{L}_{da}$), which facilitates cross-domain knowledge transfer. The former is trained with ground-truth labels, enabling the model to learn reliable evidence for known classes while expressing proper “unknownness’’ through the uncertainty mass when encountering out-of-distribution samples.
>
> For the latter one, given that cross-domain knowledge transfer is a crucial component in OSGDA, **we aim to select samples with minimal noise and to ensure that the transferred knowledge occurs with high confidence between instances of the same class across domains,** so we introduce the new evidential terms in the Evidential Adjacency-Mixup Alignment part: an interaction-level term (Int), defined as follows:
> $$
> Int = \left\|\frac{\alpha_v}{S_v}-\frac{\alpha_{v^,}}{S_{v^,}}\right\|_1
> $$
> where $\alpha = e + 1$ denotes the Dirichlet parameters derived from the evidential outputs ($e$), and $S=\sum_i \alpha_i$ is the sum of all Dirichlet parameters for an instance.
>
> The term $Int$ **captures the evidential similarity between an instance and its neighbors through the $L_1$ distance of their normalized evidence distributions.** By optimizing this term, we aim to align each instance with a confident representative of its neighborhood in evidential space, thereby facilitating cross-domain knowledge transfer.
>
> Regarding the choice of the threshold parameter $\eta$, our goal was to ensure that our ETA remains robust across diverse scenarios. To this end, we adopt a **dynamically evolving thresholding strategy**: the threshold is first randomly initialized within a predefined interval, and then gradually increased as training progresses. This dynamic scheme enables the model to incorporate a relatively large (and potentially noisy) set of unknown class samples during the early phase of training, while focusing on fewer but more reliable unknown samples in later stages. Such progression helps ETA better adapt to a wide range of OSGDA settings. **We have some discussions about this in the updated manuscript, please refer to the blue-highlighted content in Section 3.3 in the revised manuscript.**

---

> ### Author Response · Authors · 2025-11-21
>
> Q4: While fusing local and global evidence is intuitive, the paper may need clearly explain how the evidence from the two branches (Dirichlet distributions) is formally combined/integrated (e.g., using Dempster-Shafer theory's combination rule or a simpler summation/concatenation of concentration parameters) and why that specific fusion method is optimal for this graph problem.
>
> R4: Thank you for the question. In our ETA, we design a dual-branch architecture in which GCN is used to aggregate locally oriented information while PAN aggregates globally oriented information. The evidence produced by the two branches is then fused using Dempster–Shafer theory to obtain more credible and robust evidence scores. The underlying intuition is as follows:
>
> - **Complementary aggregation of local and global graph information.** The dual-branch design enables the model to jointly capture node-centric local structures and high-order global topological patterns, thereby leveraging graph topology and node attributes in a more comprehensive manner.
> - **Handling conflicting evidence under the OSGDA setting.** In OSGDA setting, the target graph may contain classes unseen in the source domain. In such cases, global aggregation may incorporate noise from out-of-distribution (OOD) nodes, potentially producing evidence that conflicts with the locally aggregated branch. To manage this, we employ the Dempster–Shafer evidence fusion rule. **The core principle of the rule is to retain only the parts where both branches provide consistent support and treat the inconsistent portions as conflict mass, which is subsequently normalized.** This process preserves consensual information while explicitly capturing disagreement as uncertainty mass, thereby yielding a more reliable representation of “unknownness.’’ Such a property aligns naturally with the requirements of OSGDA.
>
> In summary, **the dual-branch design enables richer semantic aggregation, while the evidence fusion mechanism offers a principled way to incorporate and interpret branch-level conflicts.** Together, they endow ETA with strong unknown-class awareness, making the method well suited for the OSGDA scenario.
>
> **Additionally, we have rewritten the related part of Dempster’s Rule in the updated manuscript, please refer to the blue-highlighted content in Section 2.3, Dempster’s Rule of Combination in the revised manuscript.**
>
> Thank you once again for your insightful comments. We hope that our clarifications satisfactorily address your concerns and further substantiate the effectiveness and validity of our proposed approach.

---

### Official Review · Reviewer_Ke4V · 2025-10-30

**Soundness:** 2
**Presentation:** 2
**Contribution:** 2
**Rating:** 2
**Confidence:** 4

**Summary:**

This paper investigates the challenging open-set graph domain adaptation problem, it proposes a dual evidence-aware uncertainty learning framework ETA that simultaneously identifies unknown target nodes and enhances knowledge transfer under the evidential learning theory. The proposed ETA integrates edge-oriented and path-oriented branches, generalizes evidential learning for unknown quantification, and performs cross-domain MixUp to generate virtual samples as auxiliary supervision signals for robust domain alignment.

**Strengths:**

1. The authors propose a dual evidence-aware uncertainty learning framework ETA to investigate the challenging open-set graph domain adaptation problem.
2. Extensive experiments on citation networks demonstrate that ETA significantly outperforms state-of-the-art baselines in open-set graph domain adaptation tasks.

**Weaknesses:**

1. The motivation of the article is unclear. As described in the introduction that there exists several works in the field of open-set graph domain adaptation [1,2], what is the significance of the author's proposed ETA?
2. The proposed ETA generalizes evidential learning for unknown quantification, focusing on evidential learning after classification, that is, ETA mainly addresses open-set problem, regardless of whether it is a graph-structured domain adaptation.
3. Single dataset, only focusing on citation network, and lacking blog network or airline network to verifing effectiveness of ETA.

[1] Open-set graph domain adaptation via separate domain alignment

[2] Dual structured exploration with mixup for open-set graph domain adaption

**Questions:**

Please refer to weakness

---

> ### Author Response · Authors · 2025-11-21
>
> Thanks for your thoughtful comments and valuable feedback. We'll spare no effort to address your concerns in the following.
>
> Q1: The motivation of the article is unclear. As described in the introduction that there exist several works in the field of open-set graph domain adaptation, what is the significance of the author's proposed ETA?
>
> R1: Thank you for your thoughtful comments. The open-set graph domain adaptation (OSGDA) problem is a recently emerging challenge that has been less researched. The paper [1] mentioned by the reviewer addresses *graph-level* classification, whereas we focus on *node-level* graph domain adaptation, which is quite different and **remains largely underexplored** [2].
>
> In OSGDA, a fundamental challenge lies in reliably distinguishing known classes from unknown ones, so as to mitigate their mutual interference during training. Prior work [2] predominantly relies on predictive entropy as the criterion for identifying unknown classes. However, the upper bound of entropy depends on the total number of classes, which varies considerably across different OSGDA settings. Consequently, the entropy range also fluctuates widely, making entropy-based thresholds unstable and prone to large deviations. **In contrast, the uncertainty mass derived from Evidential Deep Learning (EDL) is inherently bounded within $[0,1]$, regardless of the number of classes, offering substantially better stability.** This motivated us to introduce EDL into OSGDA with the expectation that its normalized uncertainty measure would yield more consistent behavior across diverse OSGDA scenarios, and **we conduct an ablation study** comparing ordinary predictive entropy ($\text{ETA}_{entropy}$) with uncertainty mass ($\text{ETA}$) shown as follows. The results clearly show that replacing the uncertainty mass with entropy leads to noticeable performance degradation, further validating the superiority of EDL-based uncertainty modeling in OSGDA.
>
> |Method|A2D|D2A|A3C|C2A|C2D|D2C|
> |-|-|-|-|-|-|-|
> $\text{ETA}_{entropy}$|60.32/56.63|50.83/47.66|60.33/57.76|55.54/52.28|63.38/60.28|56.64/53.36|
> $\text{ETA(Ours)}$|68.65/66.34|52.06/51.94|69.33/68.91|60.01/59.13|73.03/71.39|62.07/61.71|
>
> **Our ETA is the first** to introduce EDL to the OSGDA setting. The key motivation arises from the **conceptual alignment** between EDL and the fundamental challenge in OSGDA. EDL inherently models confidence and explicit uncertainty by quantifying the evidence supporting each class, while also assigning an explicit **“uncertainty mass”** based on Dempster–Shafer theory. This uncertainty mass reflects the absence of supporting evidence and corresponds precisely to the notion of “unknown” in open-set scenarios.
>
> Beyond this conceptual coupling, **we provide theoretical justification** for using EDL’s uncertainty mass to distinguish unknown classes in the OSGDA setting. In addition, **we analyze the evidence-related loss term ($\mathcal{L}_{ace}$)** by decomposing its components, and present three propositions demonstrating that its optimization encourages the model to produce higher evidence for known classes and higher uncertainty for unknown classes, further supporting its suitability for OSGDA.
>
> Our empirical results further demonstrate our ETA’s advantages. On top of previous methods, **ETA achieves substantial improvements (e.g., +4.04% in HS)**, validating the effectiveness of incorporating evidential modeling into the OSGDA framework.
>
> In summary, **our ETA not only advances the study of OSGDA by achieving stronger performance than existing approaches, but also introduces a conceptually novel perspective by leveraging evidential theory to explicitly model “known” versus “unknown” states at the node level**.
>
> **In addition, we rewrite some parts of the introduction in the updated manuscript, please refer to the blue-highlighted content in Section 1 in the revised manuscript.**
>
> [1] Dual structured exploration with mixup for open-set graph domain adaption. 2024 ICML.
>
> [2] Open-set graph domain adaptation via separate domain alignment. 2024 AAAI.

---

> ### Author Response · Authors · 2025-11-21
>
> Q2: The proposed ETA generalizes evidential learning for unknown quantification, focusing on evidential learning after classification, that is, ETA mainly addresses open-set problem, regardless of whether it is a graph-structured domain adaptation.
>
> R2: Thanks for the comment! **Our ETA is not merely an application of EDL, instead it is a method specifically designed for OSGDA, with a stronger emphasis on cross-domain knowledge transfer in graphs**.
>
> - In ETA, we first introduce a **dual-branch architecture (one branch based on edge-level aggregation and another on path-level aggregation)** to capture both local structural information and high-order global topology. These aggregated representations are subsequently passed through EDL to extract and fuse evidential features, enabling effective discrimination between known and unknown classes.
> - Moreover, to further combine EDL, node attributes with graph topology for cross-domain alignment, we design the **Evidential Adjacency-mixup Alignment**, which constructs cross-domain neighbor sets from both the attribute perspective and the topological perspective, leveraging evidence to discover cross-domain inter-class neighbor relations. By combining these complementary perspectives, ETA achieves robust and informative cross-domain knowledge alignment.
>
> The above modules are all graph-specific and designed in a principled and effective manner. Therefore, our ETA is a tailored solution for the OSGDA problem, rather than a straightforward extension of existing EDL techniques.
>
> Q3: Single dataset, only focusing on citation network, and lacking blog network or airline network to verify the effectiveness of ETA.
>
> R3: Thank you for your comment regarding the inclusion of the blog and airline network datasets. Following your suggestion, we conduct experiments on three widely used airport datasets[1] and two blog datasets[2], namely USA (U), Brazil (B), Europe (E) for the airport datasets (totally 4 classes) and Blog1 (B1), Blog2 (B2) for the blog datasets (totally 6 classes). For each experiment, we choose two classes as unknown classes, and the others remain known. Besides, we choose some baselines for comparison, the results are as follows:
>
> Methods|U->B|U->E|B->E|B1->B2|
> |-|-|-|-|-|
> SpegReg[3]|33.24/22.41|28.32/22.87|27.18/13.24|29.67/18.52
> SDA|49.06/40.53|42.89/33.38|45.69/35.31|52.35/43.96
> ETA(Ours)|**56.31/52.43**|**52.53/47.89**|**50.38/48.34**|**62.23/56.62**|
>
> The values in each cell indicate $ACC$/$HS$. As shown in the results, our ETA shows notable improvements on the HS compared to the baselines, demonstrating the robustness and superiority of our method.
>
> **In the updated manuscript, we have included this additional experiment in the Appendix, please refer to the blue-highlighted content in Appendix D in the revised manuscript.**
>
> [1] struc2vec: Learning node representations from structural identity. 2017 KDD.
>
> [2] Unsupervised streaming feature selection in social media. 2015 CIKM.
>
> [3] Graph domain adaptation via theory-grounded spectral regularization. 2023 ICLR.
>
> In light of the above clarifications, we hope that our responses adequately address your concerns and that you may consider revising your evaluation accordingly. Our work would improve in accordance with your constructive suggestions.

---

### Official Review · Reviewer_4Wwf · 2025-10-31

**Soundness:** 4
**Presentation:** 3
**Contribution:** 3
**Rating:** 8
**Confidence:** 4

**Summary:**

This paper proposes a dual evidence-aware uncertainty learning framework named ETA to addresses the open-set graph domain adaptation problem. ETA uses a dual-branch encoder to extract node features, leverages evidential learning to quantify uncertainty and identify unknown nodes, and performs cross-domain Mixup with evidential adjacency-consistent uncertainty for robust domain alignment. Experiments on three citation network datasets show ETA outperforms state-of-the-art baselines in classifying known target nodes and detecting unknown ones.

**Strengths:**

**Originality:** This work pioneers the use of evidential deep learning and Dempster's rule in open-set GDA, tackling the underexplored tasks of uncertainty quantification and unknown class detection.

**Quality:** The proposed framework is technically sound and well-developed. The authors provide theoretical support for the core mechanism of uncertainty detection, providing a solid foundation for the efficacy of the model. The experimental section is comprehensive, the t-SNE visualizations and uncertainty analysis are particularly compelling.

**Clarity:** The paper is well organized and easy to understand. The model architecture diagram intuitively illustrates the composition and data flow of ETA, so that the reader can quickly grasp the core idea of the model.

**Significance:** The studied problem breaks through the limitations of traditional GDA and makes it more suitable for practical applications where new categories or unknown entities often appear.

**Weaknesses:**

**About the case about the problem:** Although the studied problem is novel, there is a lack of practical cases to illustrate its practical significance.

**About the Hyperparameters:** The model introduces several key hyperparameters, notably the uncertainty threshold $\eta$. In a fully unsupervised target domain scenario, it is not clear how an optimal $\eta$ could be set a priori.

**About the Computational Complexity:** The cross-domain neighbor search and MixUp may introduce significant computational overhead.

**Questions:**

1. Could you give an example of open-set GDA in real life, so that people can understand its true meaning more clearly.
2. Could you elaborate on how to select an appropriate threshold $\eta$ when facing different target domains?
3. What strategies does ETA employ to handle the additional computational overhead introduced during the domain alignment process?

---

> ### Author Response · Authors · 2025-11-21
>
> We are grateful for your time taken to review our work and valuable feedback. We'll try our best to address your concerns in the following.
>
> Q1: Could you give an example of open-set GDA in real life, so that people can understand its true meaning more clearly.
>
> R1: Thanks for the comments! Let's consider the following concrete example in **social network analysis**: Imagine a social network where nodes represent users and edges represent connections. In a closed-set GDA scenario, a model trained on one social platform (source) to predict user interests on another (target) would assume all user interests (classes) are predefined and known. However, real-world social networks are dynamic, with new trends, subcultures, or emerging topics (e.g., a new cryptocurrency, a niche hobby, a viral challenge) constantly appearing. A closed-set model would likely misclassify users interested in these unknown emerging topics into existing categories, or simply fail to recognize their distinct interests. An OSGDA model, like our ETA, could identify users participating in these novel, emerging topics as belonging to an "unknown" class, rather than forcing them into an ill-fitting known category. This capability is critical for applications such as targeted advertising, trend detection, or anomaly identification. The example illustrates that **the "unknown" class in OSGDA is not merely a theoretical construct but a critical practical challenge in dynamic, real-world graph environments.** The ability of OSGDA to identify these novel instances provides a unique and valuable solution that closed-set methods cannot offer.
>
> Q2: Could you elaborate on how to select an appropriate threshold when facing different target domains?
>
> R2: Thanks for the comment! In the implementation of our ETA, we employ a **dynamically evolving thresholding strategy**: the threshold is first randomly initialized within a predefined interval, and then gradually increased as training progresses. This dynamic scheme enables the model to incorporate a relatively large (and potentially noisy) set of unknown class samples during the early phase of training, while focusing on fewer but more reliable unknown samples in later stages. Such progression helps ETA better adapt to a wide range of OSGDA settings. This strategy is also suitable for the new different target domains, as the dynamical threshold allows the model to focus more on collecting knowledge at the beginning of the training and then turns to focus on the high-confidence less-noisy knowledge, making the model more robust to new domains.
>
> **We have some discussions about this in the updated manuscript, please refer to the blue-highlighted content in Section 3.3, ABLATION STUDY AND SENSITIVITY ANALYSIS in the revised manuscript.**
>
> Q3: What strategies does ETA employ to handle the additional computational overhead introduced during the domain alignment process?
>
> R3: Thanks for the comment! During the domain alignment process, the construction of cross-domain neighbors could introduce additional computational overhead, which could make the time complexity to $\mathcal{O}(N^sN^t)$, $N$ denotes the number of the nodes of the graph domain. However, we select anchor nodes from the source domain for cross-domain alignment, which not only reduces the potential noise but also optimizes the time complexity to $\mathcal{O}(c\cdot N^t)$, $c$ denotes the number of anchor nodes, which is usually less more than $N^s$.
>
> Thank you once again for your insightful comments. We hope that our clarifications satisfactorily address your concerns and further substantiate the effectiveness and validity of our proposed approach.

---

### Official Review · Reviewer_qUfv · 2025-10-31

**Soundness:** 3
**Presentation:** 3
**Contribution:** 2
**Rating:** 4
**Confidence:** 3

**Summary:**

This work analyzes the challenges of Open-Set Graph Domain Adaptation (OSGDA), where models must transfer knowledge learned on source graphs to target graphs containing unknown classes. The authors propose a novel framework named ETA, whose core lies in leveraging evidence-based deep learning to quantify prediction uncertainty, thereby enabling principled identification of unknown-class nodes. Additionally, the paper introduces a dual-branch encoder and a novel cross-domain MixUp strategy. Experimental results demonstrate that this approach significantly outperforms existing methods across three benchmark datasets.

**Strengths:**

The core idea of this work—leveraging evidence learning to quantify uncertainty for identifying unknown classes—is innovative. It employs evidence-based deep learning to learn supporting evidence for each category and utilizes Dirichlet distributions to model class probabilities. The method is well-designed and supported by rigorous experiments.

**Weaknesses:**

This approach appears to be a clever integration of existing mature techniques (GNNs, EDL, MixUp) rather than a fundamental paradigm shift.
The main text omits discussions on computational complexity and the selection strategy for the critical hyperparameter $\eta$, relegating these crucial experimental analyses to the appendix.
Experiments are confined to homogenized citation network datasets, which are limited in variety, and the number of nodes in each dataset is not clearly specified.

**Questions:**

This approach ingeniously integrates three established techniques: GNN, EDL, and MixUp. Beyond the ultimate empirical performance gains, does this particular combination yield any new theoretical or mechanistic insights?
The design of Equation (14) appears heuristic. Beyond empirical success, is there a more comprehensive description demonstrating this as a sound form for achieving evidence consistency?
Experiments are entirely confined to citation networks. How do you assess the method's generalization potential on graphs with vastly different topologies?
What are your thoughts regarding future work?

---

> ### Author Response · Authors · 2025-11-21
>
> We are grateful for your time taken to review our work and thoughtful feedback. We'll try our best to address your concerns in the following.
>
> Q1: This approach ingeniously integrates three established techniques: GNN, EDL, and MixUp. Beyond the ultimate empirical performance gains, does this particular combination yield any new theoretical or mechanistic insights?
>
> R1: Thank you for the question. In OSGDA, a fundamental challenge lies in reliably distinguishing known classes from unknown ones, so as to mitigate their mutual interference during training. Prior works [1] predominantly rely on predictive entropy as the criterion for identifying unknown classes. However, the upper bound of entropy depends on the total number of classes, which varies considerably across different OSGDA settings. Consequently, the entropy range also fluctuates widely, making entropy-based thresholds unstable and prone to large deviations. **In contrast, the uncertainty mass derived from EDL is inherently bounded within $[0,1]$, regardless of the number of classes, offering substantially better stability.** This motivated us to introduce EDL into OSGDA with the expectation that its normalized uncertainty measure would yield more consistent behavior across diverse OSGDA scenarios. **The additional ablation study** comparing ordinary predictive entropy ($\text{ETA}_{entropy}$) with uncertainty mass ($\text{ETA}$) shown as follows further validates the superiority of EDL-based uncertainty modeling in OSGDA.
>
> |Method|A2D|D2A|A3C|C2A|C2D|D2C|
> |-|-|-|-|-|-|-|
> $\text{ETA}_{entropy}$|60.32/56.63|50.83/47.66|60.33/57.76|55.54/52.28|63.38/60.28|56.64/53.36|
> $\text{ETA}$|68.65/66.34|52.06/51.94|69.33/68.91|60.01/59.13|73.03/71.39|62.07/61.71|
>
> At the same time, graph-structured data exhibits unique topological properties. **To effectively capture both local and global structural information,** we adopt a dual-branch architecture and employ Dempster–Shafer theory to fuse evidence from different branches within the EDL framework. This enables our model to generate more reliable evidence values. Building upon these evidence values, we further perform cross-domain alignment, allowing us not only to separate known and unknown classes but also to transfer knowledge across domains, which addresses both core aspects of OSGDA.
>
> Moreover, In our work, **we provide theoretical justification** for using EDL’s uncertainty mass to distinguish unknown classes in the OS-GDA setting. Specifically, **Theorem 1** shows that in the latent representation space of EDL, the uncertainty mass of an unknown-class sample has a provable lower bound that increases as the sample moves farther away from known-class regions. This result establishes a solid theoretical basis for using uncertainty mass as a thresholding criterion for unknown-class detection. In addition, **we analyze the evidence-related loss term ($\mathcal{L}_{ace}$)** by decomposing its components, and present three propositions demonstrating that its optimization encourages the model to produce higher evidence for known classes and higher uncertainty for unknown classes, further supporting its suitability for OSGDA.
>
> In summary, **our work introduces a novel mechanism for the first time to bring EDL into the OSGDA and provides theoretical justification from both the latent representation perspective and the loss function perspective.**
>
> **In addition, we rewrite some parts of the introduction and include the ablation study in the updated manuscript, please refer to the blue-highlighted content in Section 1 and Appendix C.2 in the revised manuscript.**
>
> [1]Open-set graph domain adaptation via separate domain alignment. 2024 AAAI.

---

> ### Author Response · Authors · 2025-11-21
>
> Q2: The design of Equation (14) appears heuristic. Beyond empirical success, is there a more comprehensive description demonstrating this as a sound form for achieving evidence consistency?
>
> R2: Thank you for the question. Given that cross-domain knowledge transfer is a crucial component in OSGDA, **we aim to select samples with minimal noise and to ensure that the transferred knowledge occurs with high confidence between instances of the same class across domains,** so we introduce two new evidential terms in the Evidential Adjacency-Mixup Alignment part: an individual-level term (Ind) and an interaction-level term (Int), defined as follows:
> $$
> Ind = log(\frac{S_{v^,}-\max_k(\alpha_{v^,})}{\max_k(\alpha_{v^,})}),
> Int = \left\|\frac{\alpha_v}{S_v}-\frac{\alpha_{v^,}}{S_{v^,}}\right\|_1
> $$
> where $\alpha = e + 1$ denotes the Dirichlet parameters derived from the evidential outputs ($e$), and $S=\sum_i \alpha_i$ is the sum of all Dirichlet parameters for an instance.
>
> The first term $Ind$ **reflects the confidence level encoded in an instance’s own evidence distribution.** Optimizing $Ind$ encourages an instance not only to produce correct evidence for its class but also to output sufficiently confident evidence, thereby reinforcing the reliability of its class assignment. The second term $Int$ **captures the evidential similarity between an instance and its neighbors through the $L_1$ distance of their normalized evidence distributions.** By jointly optimizing these two terms, we aim to align each instance with a confident representative of its neighborhood in evidential space, thereby facilitating cross-domain knowledge transfer.
>
> **Moreover, we rewrite some parts of the context of equation (14), please refer to the blue-highlighted content in Section 2.4 in the revised manuscript.**

---

> ### Author Response · Authors · 2025-11-21
>
> Q3: Experiments are entirely confined to citation networks. How do you assess the method's generalization potential on graphs with vastly different topologies?
>
> R3: Thank you for the insightful comment. First we extend our ETA to airport datasets[1] and blog datasets[2], namely USA (U), Brazil (B), Europe (E) for the airport datasets (totally 4 classes) and Blog1 (B1), Blog2 (B2) for the blog datasets (totally 6 classes). For each experiment, we choose two classes as unknown classes, and the others remain known. Besides, we choose some baselines for comparison, the results are as follows:
>
> Methods|U->B|U->E|B->E|B1->B2|
> |-|-|-|-|-|
> SpegReg[3]|33.24/22.41|28.32/22.87|27.18/13.24|29.67/18.52
> SDA|49.06/40.53|42.89/33.38|45.69/35.31|52.35/43.96
> ETA(Ours)|**56.31/52.43**|**52.53/47.89**|**50.38/48.34**|**62.23/56.62**|
>
> The values in each cell indicate $ACC$/$HS$. As shown in the results, our ETA shows notable improvements on the HS compared to the baselines, demonstrating the robustness and superiority of our method.
>
> Besides, **we explore ETA in the heterogeneous graphs.** In ETA, only the backbone relies on a homogeneous GNN for graph aggregation. The remaining modules are independent of the homogeneous graph assumption. Thus, given appropriate graph representations (by simply changing the homogeneous backbone to a heterogeneous one), **ETA can in principle be applied to heterogeneous graphs as well and address OSGDA in that setting.** To further examine ETA’s potential on heterogeneous graphs, we conduct experiments on the public heterogeneous dataset **OGB-MAG** [4]. We split the graph into a source domain (pre-2017, $\text{MAG}\_{pre}$) and a target domain (post-2019, $\text{MAG}\_{post}$), and treat 20% of the classes as unknown. We compare SDA [5], UAGA [6], ETA with GCN backbone ($\text{ETA}\_{GCN}$), and ETA with a heterogeneous backbone R-GCN [7] ($\text{ETA}\_{R\text{-}GCN}$).
>
> |Method|$\text{MAG}\_{pre}\to \text{MAG}\_{post}$|
> |-|-|
> SDA|22.64/10.56|
> UAGA|23.82/12.38|
> $\text{ETA}\_{GCN}$|**30.56/23.85**|
> $\text{ETA}\_{R-GCN}$|**40.26/35.42**|
>
> The values in each cell indicate $ACC$/$HS$. SDA and UAGA perform poorly, particularly in HS, due to the scale and heterogeneity of OGB-MAG. $\text{ETA}\_{GCN}$ surpasses both baselines but still exhibits moderate HS, confirming that heterogeneity introduces additional noise that complicates distinguishing known from unknown categories. After replacing the backbone with the heterogeneous R-GCN, $\text{ETA}\_{R\text{-}GCN}$ achieves substantial improvements, especially in HS, which highlights ETA’s strong potential on heterogeneous graphs when equipped with an appropriate heterogeneous encoder.
>
> In summary, **our ETA is a generalized framework that can be extended to multiple datasets, regardless of whether the datasets are vastly different in topologies or are heterogeneous.**
>
> **We have included part of the additional experiments in the updated manuscript, please refer to the blue-highlighted content in Appendix D in the revised version.**
>
> [1] struc2vec: Learning node representations from structural identity. 2017 KDD.
>
> [2] Unsupervised streaming feature selection in social media. 2015 CIKM.
>
> [3] Graph domain adaptation via theory-grounded spectral regularization. 2023 ICLR.
>
> [4] Open graph benchmark: Datasets for machine learning on graphs. 2023 NeurIPS.
>
> [5] Open-set graph domain adaptation via separate domain alignment. 2024 AAAI.
>
> [6] Open-Set Cross-Network Node Classification via Unknown-Excluded Adversarial Graph Domain Alignment. 2025 AAAI.
>
> [7] Modeling relational data with graph convolutional networks. 2018 ESWC.

---

> ### Author Response · Authors · 2025-11-21
>
> Q4: What are your thoughts regarding future work?
>
> R4: Thank you for the question! Beyond our current research, we plan to extend our ETA to more challenging learning paradigms, such as class-imbalanced learning, label-noised learning and multi-modal learning. Furthermore, we anticipate exploring ETA's application in specific dynamic graph systems such as cybersecurity intrusion detection. Usually in this area, attackers constantly evolve their strategies. This presents a quintessential OSGDA problem, where the target domain contains 'unknown' attack types that must be accurately identified rather than misclassified, such networks often exhibit subtle topological patterns (e.g., cyclic paths). ETA’s dual-branch encoder comprehensively integrates local edge consistency and global path semantics, enabling it to uncover structural anomalies that purely feature-based methods might overlook. Besides, ETA’s evidence-aware uncertainty quantification offers a principled mechanism to screen high-risk out-of-distribution nodes for human audit. This capability helps mitigate the interpretability limitations inherent in 'black-box' predictions. Applying ETA to these domains would make a significant contribution to the advancement of trustworthy AI in security applications.
>
> **We have some brief discussion of future work in the updated manuscript, please refer to the blue-highlighted content in Section 4 in the revised manuscript.**
>
> Q5: The main text omits discussions on computational complexity and the selection strategy for the critical hyperparameter $\eta$, relegating these crucial experimental analyses to the appendix.
>
> R5: Thank you for the valuable comment. We has moved the time complexity analysis to the main text in the revised manuscript, **please refer to the blue-highlighted content in Section 2, 2.5 Time Complexity in the revised manuscript.**
>
> Regarding the choice of the threshold parameter $\eta$, our goal was to ensure that our ETA remains robust across diverse scenarios. To this end, we adopt a **dynamically evolving thresholding strategy**: the threshold is first randomly initialized within a predefined interval, and then gradually increased as training progresses. This dynamic scheme enables the model to incorporate a relatively large (and potentially noisy) set of unknown class samples during the early phase of training, while focusing on fewer but more reliable unknown samples in later stages. Such progression helps ETA better adapt to a wide range of OSGDA settings. **We have some discussions about this in the updated manuscript, please refer to the blue-highlighted content in Section 3.3, ABLATION STUDY AND SENSITIVITY ANALYSIS in the revised manuscript.**
>
> Q6: The number of nodes in each dataset is not clearly specified
>
> R6: Thanks for comments! We follow the previous works[1,2], in which citation networks are widely used benchmark datasets. The discrepancy mainly comes from the shifts caused by time, and the detailed information of the datasets are shown in following, **and we have included these information in the revised version, please refer to the blue-highlighted content in Appendix F in the revised manuscript.**
> |Datasets|Nodes|Edges|Attributes|
> |-|-|-|-|
> ACMv9|9360|15602|5571|
> Citationv1|8935|15113|5379|
> DBLPv7|5484|8130|4412
>
> [1] Pairwise Alignment Improves Graph Domain Adaptation. 2024 ICML.
>
> [2] On the Benefits of Attribute-Driven Graph Domain Adaptation. 2025 ICLR.
>
> In light of the above clarifications, we hope that our responses adequately address your concerns and that you may consider revising your evaluation accordingly. Our work would get improved in accordance with your constructive suggestions.

---

### Note · Authors · 2025-11-24

I have read and agree with the venue's withdrawal policy on behalf of myself and my co-authors.